# Dense Associative Memory on the Bures-Wasserstein Space

## Abstract

Dense associative memories (DAMs) store and retrieve patterns via energy-functional fixed points, but existing models are limited to vector representations. We extend DAMs to probability distributions equipped with the 2-Wasserstein distance, focusing mainly on the Bures–Wasserstein class of Gaussian densities. Our framework defines a log-sum-exp energy over stored distributions and a retrieval dynamics aggregating optimal transport maps in a Gibbs-weighted manner. Stationary points correspond to self-consistent Wasserstein barycenters, generalizing classical DAM fixed points. We prove exponential storage capacity, provide quantitative retrieval guarantees under Wasserstein perturbations, and validate the model on synthetic and real-world distributional tasks. This work elevates associative memory from vectors to full distributions, bridging classical DAMs with modern generative modeling and enabling distributional storage and retrieval in memory-augmented learning.

## 1 Introduction

Associative memories are a foundational paradigm for robust storage and retrieval of structured information. Classical models, such as the Hopfield network (Hopfield, 1982a) and its modern high-capacity extensions (Krotov & Hopfield, 2016; Ramsauer et al., 2020), demonstrate that high-dimensional patterns can be stored in distributed representations and retrieved accurately under partial or corrupted queries. These models formalize the principle that memory retrieval can be framed as a dynamical system evolving toward energy minima corresponding to stored patterns.

While most prior work on associative memory has focused on *vector-valued* data, many modern applications involve *probability distributions* as the fundamental objects. In representation learning, embeddings are increasingly modeled as distributions to capture uncertainty and multi-modality, requiring retrieval and manipulation at the distributional level rather than through point estimates. Gaussian embeddings in particular provide a versatile framework, beginning with Vilnis & McCallum (2014), who modeled words as multivariate Gaussians to encode semantic uncertainty and asymmetry. This paradigm has since been extended to networks and graphs via Graph2Gauss (Bojchevski & Günnemann, 2018), to documents in both unsupervised settings (Banerjee et al., 2017) and linked-document frameworks like GELD (He et al., 2020), and to richer semantic representations such as Wasserstein Gaussian embeddings (Athiwaratkun et al., 2018), Gaussian mixture embeddings (Athiwaratkun et al., 2018), and conceptualized Gaussian embeddings (Wang et al., 2021). Recent advances also include Gaussian graph neural networks for large ontologies (Wang et al., 2025) and sentence embeddings as Gaussian distributions (Yoda et al., 2024). Across these domains, Gaussian representations unify uncertainty, hierarchical structure, and distributional similarity, underscoring their versatility for modern machine learning.

More generally, in uncertainty-aware generative modeling, where probabilistic generative models such as variational autoencoders (Kingma & Welling, 2013), normalizing flows (Rezende & Mohamed, 2015), and diffusion models (Sohl-Dickstein et al., 2015; Ho et al., 2020) learn distributions over complex modalities including images, text, and 3D point-clouds. Similarly, in Bayesian inference, posterior beliefs about latent variables are encoded as probability densities (often Gaussian or Gaussian mixtures) where updating or recalling these beliefs corresponds to operations directly on distributions (Bernardo & Smith, 2009; Khan & Rue, 2023). In these contexts, it is natural to

treat entire distributions, rather than individual samples, as primary computational units. These considerations motivate the following question:

> *Can associative memories be generalized to store and retrieve probability distributions, rather than deterministic vectors?*

We address this question in the setting of Gaussian distributions. Let $\mathcal{N}(\mu_i, \Sigma_i)$, $i = 1, \ldots, N$, be the target distributions. Endowed with the *Bures–Wasserstein geometry* (Asuka, 2011; Lambert et al., 2022; Diao et al., 2023), Gaussians inherit a Riemannian structure from the Wasserstein-2 distance that captures both mean and covariance, providing a natural notion of similarity. Our goal is to design a *dense associative memory (DAM)* that robustly stores $\{\mathcal{N}(\mu_i, \Sigma_i)\}_{i=1}^{N}$ and retrieves the correct distribution from noisy or partial queries, thus extending classical DAMs from $\mathbb{R}^d$ to the non-Euclidean space of probability measures while retaining high capacity and robust retrieval. Towards that, we make the following **contributions** in this work:

1. **Wasserstein LSE energy functional.** We propose a novel energy formulation defined directly on the (Bures)-Wasserstein space, generalizing classical DAM energies to probability densities (see Section 2).
2. **Exponential storage capacity.** We prove that our model achieves storage capacity exponential in the dimensionality of the ambient space, extending classical vectorial results to the Gaussian distribution setting (see Theorem 1).
3. **Retrieval guarantees.** We establish bounds on the fidelity of retrieval under noisy query distributions, providing explicit dependence on the Wasserstein distance between stored and perturbed densities (see Section 3.2).
4. **Empirical validation.** Through experiments on synthetic Gaussian datasets and real-data experiments on probabilistic word embeddings, we demonstrate that the proposed model achieves accurate retrieval and exhibits robustness predicted by our theoretical analysis. (see Section 4)

By extending dense associative memories from vectors to probability densities, our work lays a foundation for *distributional memory architectures* in generative AI. Such memories can store, recall, and manipulate probabilistic objects, enabling memory-augmented probabilistic reasoning and uncertainty-aware generative computation.

**Notation and definitions.** For a positive integer $N$, let $[N] := \{1, 2, \ldots, N\}$. We write $\|\cdot\|$ for the Euclidean norm and $\langle \cdot, \cdot \rangle_{L^2}$ for the $L^2$ inner product between probability measures. Throughout, we work in the space $\mathcal{P}_2(\mathbb{R}^d)$ of probability measures with finite second moment, equipped with the 2-Wasserstein distance $W_2(\mu, \nu) = \inf_{\gamma \in \Gamma(\mu, \nu)} \int_{\mathbb{R}^d \times \mathbb{R}^d} \|x - y\|^2 \, d\gamma(x, y)$, where $\Gamma(\mu, \nu)$ is the set of couplings with marginals $\mu, \nu$. For a functional $\mathcal{F} : \mathcal{P}_2(\mathbb{R}^d) \to \mathbb{R}$, its first variation at $\mu$ is $\delta_\mu \mathcal{F}(\mu)(x)$, defined via $\frac{d}{d\varepsilon} \mathcal{F}(\mu + \varepsilon(\mu' - \mu))\big|_{\varepsilon=0} = \int \delta_\mu \mathcal{F}(\mu)(x) \, (\mu' - \mu)(x) \, dx$. The Wasserstein gradient is $\nabla_{\mathcal{W}} \mathcal{F}(\mu)(x) := \nabla_x \delta_\mu \mathcal{F}(x, \mu)$, and the associated (negative) gradient flow is the continuity equation $\partial_t \mu_t + \nabla \cdot (\mu_t v_t) = 0$, $v_t(x) = -\nabla_{\mathcal{W}} \mathcal{F}(\mu_t)(x)$, often written $\dot{\mu}_t = -\nabla_{\mathcal{W}} \mathcal{F}(\mu_t)$. For a measurable $T : X \to Y$ and $\mu$ on $X$, the push-forward is $T_{\#}\mu(B) := \mu(T^{-1}(B))$ for measurable $B \subseteq Y$.

The squared Wasserstein distance between two Gaussians is given by the Bures metric $W_2^2(\mathcal{N}(\mu_1, \Sigma_1), \mathcal{N}(\mu_2, \Sigma_2)) = \|\mu_1 - \mu_2\|^2 + \mathrm{Tr}(\Sigma_1 + \Sigma_2 - 2(\Sigma_1^{1/2} \Sigma_2 \Sigma_1^{1/2})^{1/2})$. The Bures–Wasserstein gradient at $\mathcal{N}(\mu, \Sigma)$ becomes the projection of the Wasserstein gradient to the tangent space at $\mathcal{N}(\mu, \Sigma)$ (Lambert et al., 2022, page 22) and it further reduces to finite-dimensional gradients: $\nabla_{\mathcal{W}} \mathcal{F}(m, \Sigma) = (\nabla_m \mathcal{F}(m, \Sigma), \nabla_\Sigma \mathcal{F}(m, \Sigma))$. Moreover, if $X \sim \mathcal{N}(m_0, \Sigma_0)$ and $Y \sim \mathcal{N}(m_1, \Sigma_1)$ with $\Sigma_0, \Sigma_1 \in \mathbb{S}_{++}^d$, the unique optimal transport map $T : \mathbb{R}^d \to \mathbb{R}^d$ pushing $X$ to $Y$ under quadratic cost is affine: $T(x) = m_1 + A(x - m_0)$, where $A = \Sigma_0^{-1/2}(\Sigma_0^{1/2} \Sigma_1 \Sigma_0^{1/2})^{1/2} \Sigma_0^{-1/2}$. See Ambrosio et al. (2005); Asuka (2011); Lambert et al. (2022) for additional details.

## 2 ENERGY FUNCTIONALS IN THE WASSERSTEIN SPACE

The key design choice in associative memories lies in specifying the energy function that drives retrieval dynamics. Classical Hopfield networks use a quadratic energy, yielding only $O(d)$ storage capacity in dimension $d$. By contrast, the log-sum-exp (LSE) energy of Krotov & Hopfield (2016)

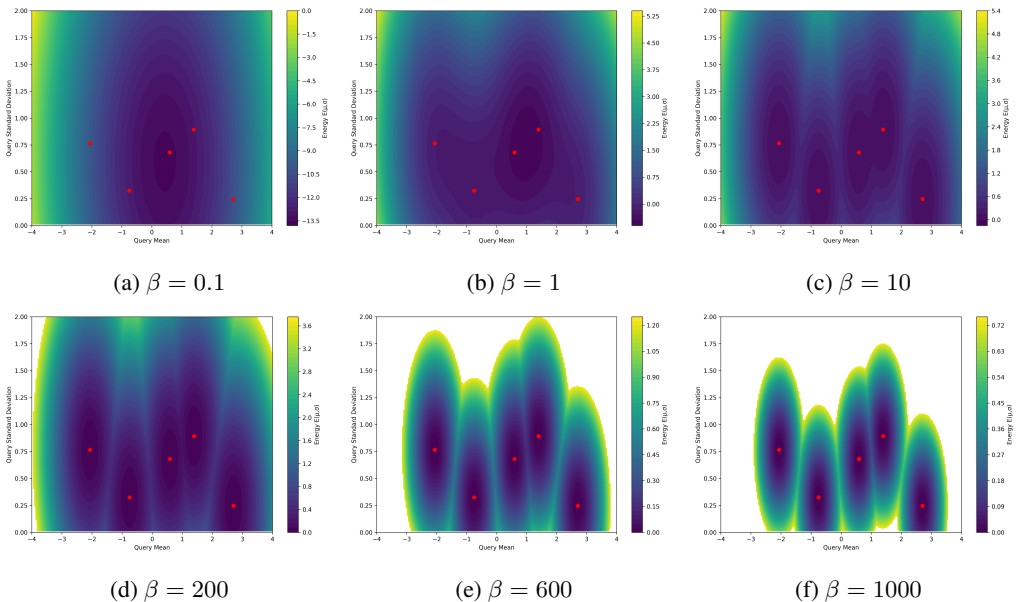

Figure 1: Energy landscape $E(\xi)$ (Equation 1) for query one-dimensional Gaussians $\xi = \mathcal{N}(\mu, \sigma^2)$ evaluated on a $200 \times 200$ grid with $\mu$ uniformly spaced in $[-4, 4]$ and $\sigma$ uniformly spaced in $[0.01, 2]$. Red dots indicate $N = 5$ one-dimensional Gaussian measures $X_i$ sampled uniformly at random with means in the interval $[-3, 3]$ and standard deviations in $[0.2, 1.0]$. The temperature parameter $\beta$ varies from $0.1$ to $1000$ across subfigures.

introduces a sharper nonlinearity and dramatically improves efficiency. For a query $\xi \in \mathbb{R}^d$ and stored patterns $\{X_i\}_{i=1}^N$, the energy is $E(\xi) = -\beta^{-1} \log(\sum_{i=1}^N \exp(-\beta\|X_i - \xi\|^2))$, with temperature parameter $\beta > 0$. As $\beta \to \infty$, this approaches the negative maximum similarity, yielding a smooth approximation to hard maximum retrieval. The induced energy landscape produces well-separated attractor basins, supports exponential storage capacity, and admits a probabilistic view where retrieval corresponds to Gibbs-type aggregation with weights exponentially concentrated on the nearest stored pattern.

Extending associative memory to probability distributions requires replacing Euclidean distance with a suitable similarity measure. We work in the Wasserstein space $(\mathcal{P}_2(\mathbb{R}^d), W_2)$ and define the Log-Sum-Exp (LSE) energy for stored patterns $X_1, \ldots, X_N \in \mathcal{P}_2(\mathbb{R}^d)$ and query $\xi \in \mathcal{P}_2(\mathbb{R}^d)$ as

$$E(\xi) := -\frac{1}{\beta} \log \Big( \sum_{i=1}^N \exp\big( -\beta W_2^2(X_i, \xi) \big) \Big). \tag{1}$$

As $\beta \to \infty$, this reduces to the negative minimum Wasserstein distance, implementing a soft-min retrieval rule with a clear probabilistic interpretation: stored distributions are weighted by their Wasserstein proximity to the query. Importantly, this extension preserves the exponential storage capacity of the vector case while operating in a non-Euclidean probability space, making the LSE energy a natural choice for distributional associative memory. Figure 1 shows the log-sum-exp energy equation 1 for five one-dimensional Gaussian measures $\{X_i\}_{i=1}^5$. As $\beta$ increases from $0.1$ to $1000$, the landscape evolves from nearly flat with overlapping basins to sharp, well-separated minima. For small $\beta$, discrimination between $\{X_i\}_{i=1}^5$ is weak, while large $\beta$ yields pronounced attractors. Thus, $E$ induces a multi-modal structure in the Bures–Wasserstein geometry, with each $X_i$ serving as an attractor, which is central to our definition of *storage* in Section 3.

The variational structure of our model is encoded through the Wasserstein gradient of the energy functional $E$. By direct differentiation, one obtains

$$\nabla_{\mathcal{W}} E(\xi) = 2 \sum_{i=1}^N w_i(\xi) (T_i - \mathsf{Id}), \tag{2}$$

where $T_i$ denotes the optimal transport map from $\xi$ to the stored distribution $X_i$, and

$$w_i(\xi) := \frac{\exp\big(-\beta W_2^2(X_i, \xi)\big)}{\sum_{j=1}^N \exp(-\beta W_2^2(X_j, \xi))} \, . \tag{3}$$

The weights $w_i(\xi)$ define a Gibbs-type distribution that assigns higher influence to memories closer to $\xi$ in Wasserstein distance. Thus, the gradient aggregates transport directions from $\xi$ to all stored distributions, with distant contributions exponentially suppressed. The log-sum-exp weighting introduces a smooth competitive mechanism, ensuring both robustness of recall and sensitivity to the underlying geometry of the memory ensemble. This formulation directly extends the classical log-sum-exp energy functions of dense associative memories from vectors to the Wasserstein space of probability measures.

The retrieval mechanism corresponds to finding stationary points of the energy functional $E$ in the Wasserstein geometry. Setting $\nabla_{\mathcal{W}} E(\xi_*) = 0$ yields $\sum_{i=1}^N w_i(\xi_*)(T_i - \mathsf{Id}) = 0$ or equivalently $\sum_{i=1}^N w_i(\xi_*)T_i = \mathsf{Id}$. In other words, the stationary condition can be written as

$$\left( \sum_{i=1}^N w_i(\xi_*)T_i \right)_\# \xi_* = \xi_* \, , \tag{4}$$

showing that $\xi_*$ is invariant under the weighted barycentric transport determined by the stored memories. Defining the operator $\Phi : \mathcal{P}_2(\mathbb{R}^d) \to \mathcal{P}_2(\mathbb{R}^d)$ by

$$\Phi(\xi) := \left( \sum_{i=1}^N w_i(\xi)T_i \right)_\# \xi \, , \tag{5}$$

retrieval is characterized by the fixed points of $\Phi$. In this way, memory recall is expressed as the self-consistency condition $\Phi(\xi_*) = \xi_*$, which generalizes fixed-point equations from classical associative memories to Wasserstein spaces.

Figure 2 illustrates the $\Phi$ operator in equation 5. Panel (a) and (b) shows $W_2(\xi, \Phi(\xi))$ as a heatmap: dark regions near the means of the five Gaussians mark fixed-point neighborhoods, while bright regions indicate strong transformations. Panels (c) and (d) depict the weight functions $w_i(\xi)$, with bright regions where a Gaussian $X_i$ dominates ($w_i(\xi) \approx 1$) and dark regions where other patterns are closer ($w_i(\xi) \approx 0$). Equation 4 has a clear geometric meaning: stationary distributions are precisely those invariant under the weighted barycentric transport field of the stored memories. The fixed point $\xi_*$ is a Wasserstein barycenter with self-consistent weights $w_i(\xi_*)$, ensuring retrieval identifies a distribution that balances the geometric pull of all memories. This implicit barycentric structure directly connects to generative modeling: as in energy-based models, convergence proceeds by descending an energy landscape, but here in Wasserstein space, where attractors are full probability laws satisfying barycentric invariance. Retrieval thus becomes a generative mechanism synthesizing distributions consistent with the stored ensemble, elevating associative memory from pointwise to distributional recall and aligning it with modern generative AI.

## 3 DAM in Bures-Wasserstein Space: Storage and Retrieval

We now specialize our framework to Gaussian distributions, a natural and tractable family for distributional associative memories. Gaussians admit a closed-form expression for the 2-Wasserstein distance (Lemma 1), affine optimal transport maps, and broad applicability in machine learning (Section 1). A key challenge is controlling mean and covariance errors (Lemma 3); to obtain sharp convergence guarantees, we restrict to the case where covariance matrices commute pairwise, ensuring a common eigenbasis and simplifying transport geometry. We further assume eigenvalues lie in $[\lambda_{\mathsf{min}}, \lambda_{\mathsf{max}}]$, enabling precise quantification of separation between patterns.

### 3.1 Storage Capacity

First, we prove that exponentially many Gaussian measures can be stored, with high probability, when they are sampled from a Wasserstein sphere $\mathcal{S}_R$ of radius $R$. Let $S$ be the set of Gaussian measures in $\mathbb{R}^d$ whose covariance matrices commute pairwise. The sphere $\mathcal{S}_R$ is defined as:

$$\mathcal{S}_R := \{\mathcal{N}(\mu, \Sigma) \in S : W_2(\delta_0, \mathcal{N}(\mu, \Sigma)) = R, \ \lambda_{\mathsf{min}} \le (\lambda_i)_{i=1}^d = \text{Eigenvalues}(\Sigma) \le \lambda_{\mathsf{max}}, \ \forall i\},$$

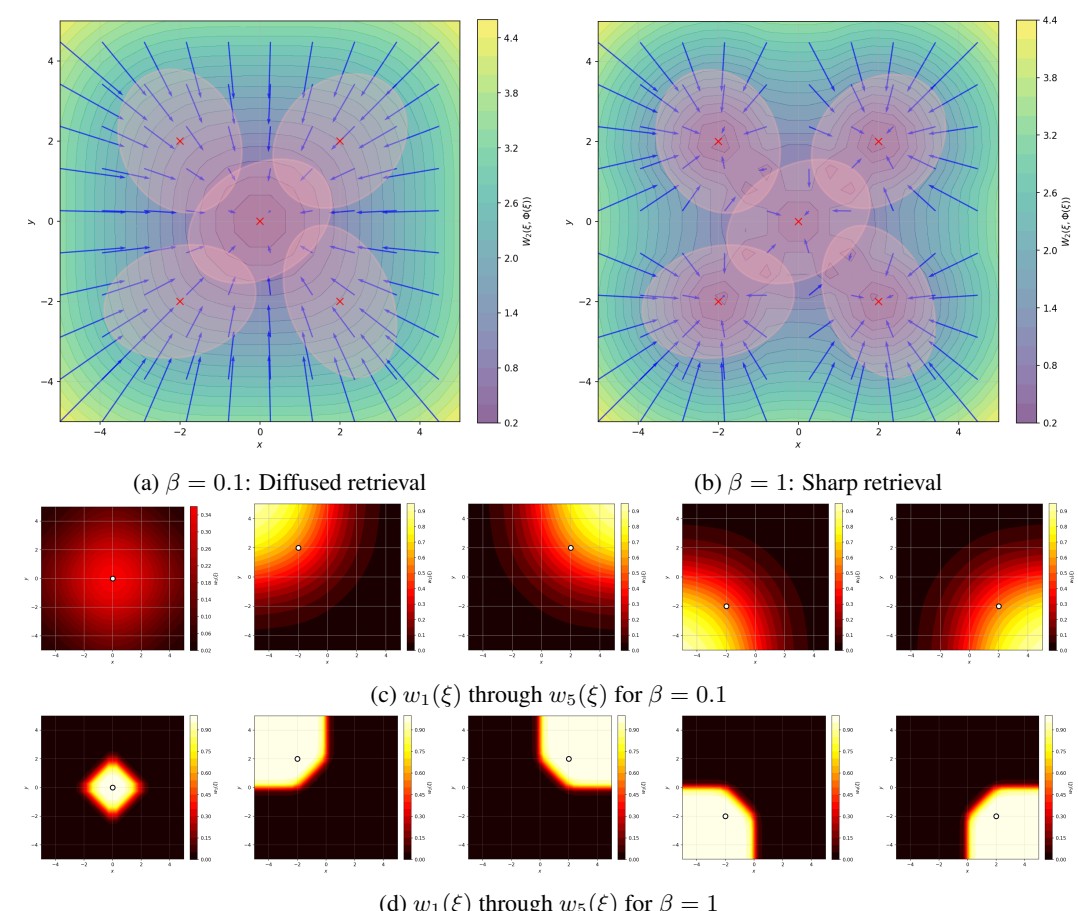

(a) $\beta = 0.1$: Diffused retrieval       (b) $\beta = 1$: Sharp retrieval

(c) $w_1(\xi)$ through $w_5(\xi)$ for $\beta = 0.1$

(d) $w_1(\xi)$ through $w_5(\xi)$ for $\beta = 1$

Figure 2: Visualization of the $\Phi$ operator and weight functions for two values of the temperature parameter $\beta$. (a,b) Heatmaps show $W_2(\xi, \Phi(\xi))$ computed on a $20 \times 20$ grid and interpolated for smooth visualization. Blue arrows indicate mean displacement vectors from $\xi$ to $\Phi(\xi)$, displayed at every second grid point. Contour lines represent level curves of constant $W_2(\xi, \Phi(\xi))$. Red ellipses show $2\sigma$ contours of stored patterns $X_1, \ldots, X_5$ with means at $(0,0), (\pm 2, \pm 2)$ and anisotropic covariances. (c,d) Weight functions $w_i(\xi)$ showing the influence of each stored pattern across the space, evaluated on the same $20 \times 20$ grid. Query distributions have fixed covariance $0.5I$. Parameters: $N = 5$, with $\beta = 0.1$ (diffused retrieval) and $\beta = 1$ (sharp retrieval).

where $0 < \lambda_{\min} < \lambda_{\max} < \infty$. We provide a practical method to perform random sampling of Gaussian measures from $\mathcal{S}_R$ in Algorithm 3, which is introduced in the proof of Theorem 1.

**Definition 1** (Storage of a Gaussian measure). *Assume that around every pattern $X_i$, a Wasserstein ball $B_i$ is given. The pattern $X_i$ is stored if there exists a unique fixed point $X_i^*$ to which all $\xi \in B_i$ converge and $B_i \cap B_j = \emptyset$ for all $i \neq j$.*

**Assumption 1** (Separation condition and beta constraint). *Let $X_i = \mathcal{N}(\mu_i, \Sigma_i)$, for $i \in [N]$, be $d$-dimensional Gaussian measures and suppose the eigenvalues of $\Sigma_i$ lie in the bounded interval $[\lambda_{\min}, \lambda_{\max}]$. Define $M_W := \max_{i \in [N]} W_2(\delta_0, X_i)$ where $\delta_0$ is the delta measure at the origin. Suppose $\beta$ is the temperature parameter from the definition of the energy functional in equation 1.*

1. *Assume the separation between Gaussian measures $\{X_i\}_{i=1}^N$ satisfies*
   $$\min_{j \neq i}(-\log\langle X_i, X_j\rangle_{L^2}) \geq \frac{d}{2}\log(4\pi\lambda_{\max}) + \frac{1}{\beta\lambda_{\min}}\log(N^3\beta(4M_W^2 + 2d(\lambda_{\max} + \lambda_{\min}))).$$

2. *Assume $\beta$ satisfies the constraint $\beta > \frac{e^2}{(4M_W^2 + 2d(\lambda_{\max} + \lambda_{\min}))N^3}$.*

**Theorem 1.** *Let $0 < p < 1$ and let $0 < \lambda_{\min} < \lambda_{\max} < \infty$. Define $\gamma := \lambda_{\max}/\lambda_{\min}$, $\alpha := 1 - 2\log(\gamma)$, $R := \sqrt{d(\lambda_{\max} + \lambda_{\min})}$. Assume $\gamma < \sqrt{e}$. Then, there exists $d_0 \in \mathbb{N}$ such that for all*

---

**Algorithm 1** One step of DAM update ($\Phi$ operator)

---

**Require:** Current state $\xi = \mathcal{N}(m, \Omega)$, stored patterns $\{X_i\}_{i=1}^N$, temperature parameter $\beta$
**Ensure:** Updated state $\xi' = \Phi(\xi) = \mathcal{N}(m', \Omega')$
 1: **Step 1: Compute Wasserstein distances to all stored patterns**
 2: **for** $i = 1$ to $N$ **do**
 3: $\quad D_i \leftarrow \|\mu_i - m\|^2 + \mathsf{tr}(\Sigma_i + \Omega - 2(\Sigma_i^{1/2}\Omega\Sigma_i^{1/2})^{1/2})$
 4: **end for**
 5: **Step 2: Compute softmax weights**
 6: **for** $i = 1$ to $N$ **do**
 7: $\quad w_i \leftarrow \frac{\exp(-\beta D_i)}{\sum_{j=1}^N \exp(-\beta D_j)}$
 8: **end for**
 9: **Step 3: Compute transport map coefficients**
10: **for** $i = 1$ to $N$ **do**
11: $\quad A_i \leftarrow \Sigma_i^{1/2}(\Sigma_i^{1/2}\Omega\Sigma_i^{1/2})^{-1/2}\Sigma_i^{1/2}$
12: **end for**
13: **Step 4: Update means and covariances**
14: $m' \leftarrow \sum_{i=1}^N w_i\mu_i$
15: $\tilde{A} \leftarrow \sum_{i=1}^N w_i A_i$
16: $\Omega' \leftarrow \tilde{A}\Omega\tilde{A}^T$
17:
18: **return** $\xi' = \mathcal{N}(m', \Omega')$

---

*$d > d_0$, we can randomly sample $N = \left\lfloor \sqrt{\frac{p}{2}} \exp\left(\frac{d\alpha^2}{16}\right) \right\rfloor$ Gaussian measures from the Wasserstein sphere $\mathcal{S}_R$ using Algorithm 3 and condition (1) in Assumption 1 is satisfied. Furthermore, if $\beta > 3\alpha/\lambda_{\min}$ and $d > d_0$ then condition (2) in Assumption 1 is satisfied. Consequently, the energy functional equation 1 has storage capacity at least $\Omega(N)$ with probability at least $1 - p$.*

Theorem 1 extends the exponential storage capacity results of Ramsauer et al. (2020)[Theorem 3] from the Euclidean space to the far more intricate Bures–Wasserstein space of Gaussian measures. While the order of the radius of the Wasserstein sphere $R$ remains $\Theta(\sqrt{d})$, the underlying analysis is considerably more delicate. In the Euclidean setting, separation can be quantified directly via inner products of vectors, i.e., $\Delta_i := x_i^\top x_i - \max_{j \neq i} x_i^\top x_j$, and independence of the coordinates makes concentration relatively straightforward. By contrast, in the Wasserstein setting we must characterize separation through the $L^2$ inner product between Gaussian measures, which depends nonlinearly on both means and covariance spectra. Establishing high-probability separation therefore requires controlling random eigenvalues of covariance matrices, which are exchangeable but not independent, and necessitates careful use of concentration inequalities in high dimensions. Moreover, unlike the Euclidean case where exponential storage capacity was proved only for a fixed $\beta$ ($\beta = 1$), our analysis establishes exponential capacity uniformly over a range of $\beta$ values, further complicating the proof. Finally, the technical constraint $\gamma < \sqrt{e}$ arises solely from bounding the joint distribution of eigenvalues, and we conjecture it can be removed with sharper concentration tools. However, we also emphasize that this conditions ensures that we dealing with actual high-dimensional Gaussian (with concentrated spectrum) rather than spectrum with decaying properties which may implicitly be lower-dimensional. Together, these challenges highlight that the extension to the Bures–Wasserstein geometry is highly non-trivial, requiring using probabilistic techniques and leveraging the geometric structure of the space more than in the Euclidean case.

## 3.2 RETRIEVAL GUARANTEES

We now focus on retrieval and introduce Algorithm 1 which can be used iteratively to retrieve the stored Gaussian distribution given a query Gaussian distribution within a Wasserstein ball around the stored Gaussian distribution.

Algorithm 1 operationalizes the Wasserstein dense associative memory update rule through a sequence of structured steps. First, it computes the pairwise Wasserstein distances between the query measure and each stored Gaussian using the Bures–Wasserstein metric. These distances are then

used to construct softmax weights that quantify the influence of each stored pattern. Next, the algorithm determines the optimal transport maps for the covariance matrices, represented by the matrices $A_i$, which specify the linear transformation required to align each stored covariance with the query. Finally, both the mean and covariance of the query measure are updated via a weighted combination of these transport maps, effectively performing a single-step retrieval towards the attractor associated with the most relevant stored pattern.

This procedure can be viewed as a natural generalization of the simple weighted averaging used in the Euclidean dense associative memory case, extended to a transport-based aggregation that respects the geometry of probability distributions. Concretely, the energy functional $E$ in equation 1 induces the following *Wasserstein gradient* at a Gaussian measure $\mu \in \mathcal{P}_2(\mathbb{R}^d)$:

$$\nabla_{\mathcal{W}} E(\xi)(x) = 2 \sum_{i=1}^{N} w_i(\xi)\big(T_i(x) - x\big) = 2 \sum_{i=1}^{N} w_i(\xi)\Big(\mu_i + A_i(x - m) - x\Big),$$

where $A_i = \Sigma_i^{1/2}(\Sigma_i^{1/2}\Omega\Sigma_i^{1/2})^{-1/2}\Sigma_i^{1/2}$ and the weights $w_i(\xi)$ are as in equation 3 which has an explicit form due to closed form availability of the Wasserstein-2 metric between Gaussians. Furthermore, as the Bures-Wasserstein gradient is the projection of the Wasserstein gradient to the tangent space at $\mu$, we can simply set the Wasserstein gradient to zero. Rather than explicitly simulating the Wasserstein gradient flow $\frac{d}{dt}\mu_t = -\nabla_{\mathcal{W}} E(\mu_t)$, Algorithm 1 solves the implicit fixed-point equation $\xi^{\text{new}} = \Phi(\xi)$ with $\Phi(\xi) = \left(\sum_{i=1}^{N} w_i(\xi)T_i\right)_{\#}\xi$. This approach is motivated by efficiency and stability: solving the implicit equation directly moves the query measure closer to a stationary point of $E$ in a single step, effectively "jumping" to the basin of attraction of the most relevant stored pattern. By contrast, an explicit discretization of the Wasserstein gradient flow may require many small steps and careful tuning of step sizes, while still potentially under-shooting or oscillating around the fixed point.

Next, we prove the rate of convergence in 2-Wasserstein distance of a query measure in the basin of attraction of a stored Gaussian measure to the fixed point in the same basin of attraction.

**Theorem 2.** *Let $\{X_i = (\mu_i, \Sigma_i)\}_{i=1}^{N}$ be Gaussian measures such that the eigenvalues of $\Sigma_i$ lie in a bounded interval $[\lambda_{\min}, \lambda_{\max}]$ and $\Sigma_i\Sigma_j = \Sigma_j\Sigma_i$ for all $i, j \in [N]$. Define $M_W := \max_i W_2(X_i, \delta_0)$ where $\delta_0$ is the delta measure at the origin. Suppose that the Assumption 1 holds. Let $r = 1/\sqrt{\beta N}$, $B_i = \{\nu \in \mathcal{P}_2(\mathbb{R}^d) : W_2(\nu, X_i < r)\}$, and suppose $\xi = \mathcal{N}(m, \Omega)$ be a query measure such that $W_2(X_i, \xi) < r$ and $\Sigma_i\Omega = \Omega\Sigma_i$ for all $i \in [N]$. Then (1) There exists a unique fixed point $X_i^* \in B_i$ and (2) For any fixed $\varepsilon < r$ we have*

$$W_2(\Phi^n(\xi), X_i^*) < \varepsilon \quad \text{for all} \quad n \geq \left\lceil \frac{\log\left(\frac{\varepsilon}{2}\sqrt{\beta N}\right)}{\log\left(\frac{144\beta M_W^2}{N}\right)} \right\rceil.$$

**Corollary 1.** *By Theorem 1 exponentially (in $d$) many Gaussian measures $N$ can be stored on a Wasserstein sphere with high probability. Hence, for any fixed $\varepsilon < r$ and large enough dimension $d$, we can sample sufficiently large number of Gaussian measures $N$ such that $W_2(\Phi(\xi), X_i^*) < \varepsilon$. In other words, Algorithm 1 converges to the fixed point $X_i^*$, starting from $\xi$, in one step.*

The next theorem obtains a quantitative bound on the one-step retrieval error.

**Theorem 3.** *Let $\{X_i\}_{i=1}^{N}$ be Gaussian measures sampled using Algorithm 3 and $\xi = \mathcal{N}(m, \Omega)$ be a query measure such that $W_2(X_i, \xi) < r$. Let $r = 1/\sqrt{\beta N}$, $B_i = \{\nu \in \mathcal{P}_2(\mathbb{R}^d) : W_2(\nu, X_i < r)\}$. Under the assumptions of Lemma 3 and 5, we obtain $W_2(\Phi(\xi), X_i) \leq 3/\sqrt{\beta N}$.*

**Corollary 2.** *By Theorem 1 exponentially (in dimension $d$) many Gaussian measures $N$ can be stored on a Wasserstein sphere with high probability. Hence, the retrieval error in one step is exponentially small in dimension $d$.*

In the Euclidean setting, Ramsauer et al. (2020)[Theorems A8, A9] show that both convergence rate and retrieval error decay exponentially with the separation parameter $\Delta_i$. Our Theorems 2 and 3 establish analogous results in the Wasserstein setting through a simplified analysis: when storing $N = \Omega(e^d)$ Gaussian measures, both the convergence rate and retrieval error decay exponentially in the dimension $d$. This demonstrates that the favorable scaling properties of dense associative memories extend naturally to distributional representations.

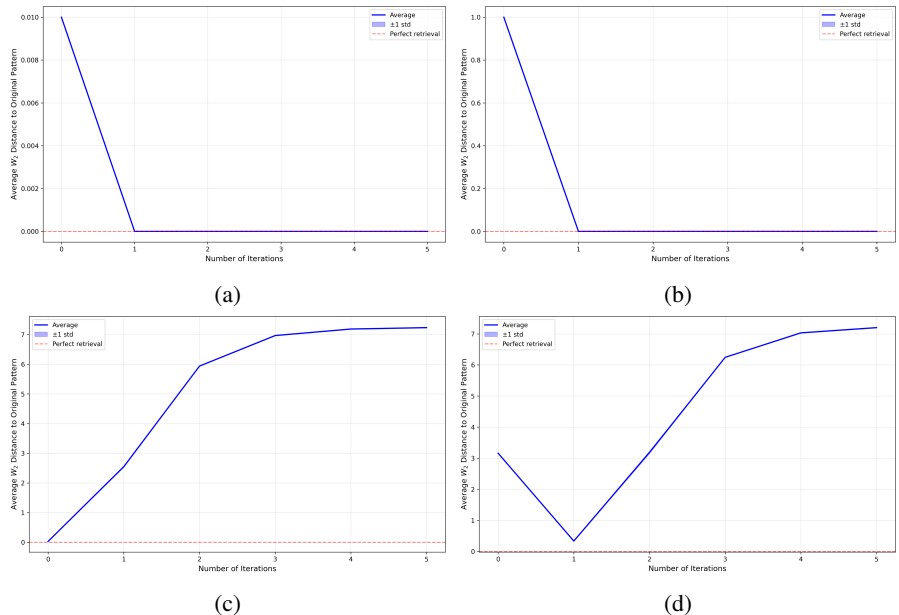

Figure 3: Convergence of retrieval dynamics (Algorithm 1) for perturbed Gaussian measures. Average 2-Wasserstein distance to original measures over iterations for different values of temperature parameter $\beta$ and perturbation radius $r$. (a) $\beta = 1, r = 1/\sqrt{\beta N}$, (b) $\beta = 1, r = 100/\sqrt{\beta N}$, (c) $\beta = 0.1, r = 1/\sqrt{\beta N}$, (d) $\beta = 0.1, r = 100/\sqrt{\beta N}$. Averages computed over 7500 perturbed measures from $N = 10000$ sampled Gaussians in dimension 50 with eigen-value bounds $\lambda_{\min} = 1, \lambda_{\max} = 1.1$. The dotted red horizontal line indicates the convergence threshold at 2-Wasserstein distance of $10^{-6}$.

# 4 NUMERICAL EXPERIMENTS

## 4.1 SYNTHETIC DATA

Figure 3 (and Figure 6 in Appendix Section 7.2.2) illustrates the convergence of our retrieval dynamics under multiple iterations of Algorithm 1, showing the average 2-Wasserstein distance to the original stored patterns. We sample $N = 10000$ Gaussian patterns on a Wasserstein sphere of radius $R = d(\lambda_{\max} + \lambda_{\min})$ with $d = 50$, $\lambda_{\min} = 1$, $\lambda_{\max} = 1.1$, using a common eigen-basis and bounded eigenvalues (sampled using Algorithm 2 in Section 7.2.1). Retrieval starts from $0.75N$ randomly perturbed patterns at 2-Wasserstein distance $r$ from their originals.

The parameter $\beta$ critically affects retrieval: for $\beta = 1$, the dynamics converge to stored patterns, exhibiting associative memory behavior, while for $\beta = 0.1$, convergence fails as the weighted barycentric transport nearly equally averages all patterns. We test two perturbation radii: $r = 1/\sqrt{\beta N}$, which aligns with the contraction guarantee from Lemma 5, and a 100-fold larger radius. For $\beta = 1$, retrieval succeeds even at the larger radius, suggesting the theoretical bound is conservative; for $\beta = 0.1$, retrieval fails at both radii. Results with non-commuting covariances are in Section 7.2.3.

### 4.1.1 REAL-WORLD DATA

To evaluate our distributional associative memory on real-world data, we employ Gaussian word embeddings learned from natural language text (Vilnis & McCallum, 2014), which represents words as multivariate Gaussian distributions. We train embeddings on the Text8 corpus, a standard benchmark containing 17 million tokens of cleaned Wikipedia text. From this corpus, we construct a vocabulary of $N = 10000$ most frequent words. The Gaussian embeddings are trained in $d = 50$ dimensions using spherical covariances (i.e., the covariance matrix $\Sigma = \sigma^2 I$ for some $\sigma$), where each word is represented as $\mathcal{N}(\mu, \sigma^2 I)$ with $\mu \in \mathbb{R}^{50}$, $\sigma \in \mathbb{R}_+$. Training in Vilnis & McCallum (2014) employs a 5-epoch schedule using KL divergence as the energy function.

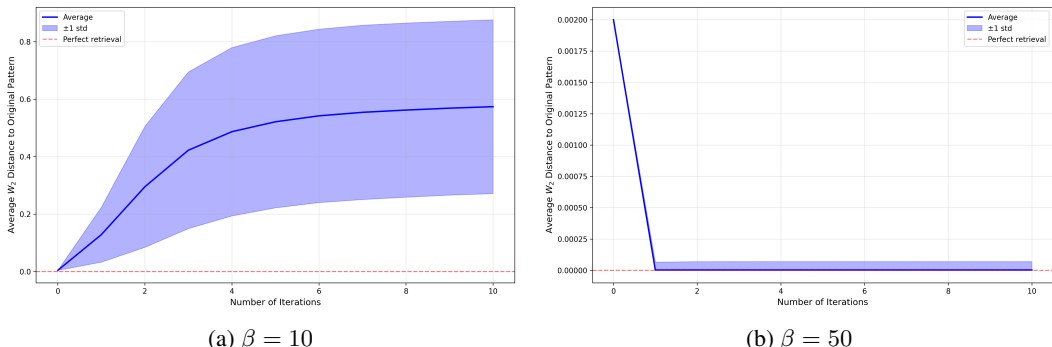

(a) $\beta = 10$                  (b) $\beta = 50$

Figure 5: Convergence of retrieval dynamics (Algorithm 1) for perturbed Gaussian measures with different temperature parameters. Average 2-Wasserstein distance to original measures over iterations. Averages computed over 3750 perturbed measures from $N = 10000$ Gaussian measures in dimension 50. The shaded regions represent $\pm 1$ standard deviation. (a) $\beta = 10$: no convergence (b) $\beta = 50$: convergence with less variance.

To test retrieval dynamics, we randomly select 5 words from $N = 10000$ words. For each selected word with Gaussian representation $\mathcal{N}(\mu_i, \sigma_i^2 I)$, we generate a perturbed query by changing both mean and variance to achieve a 2-Wasserstein distance of $1/\sqrt{\beta N}$ from the original, where $\beta$ is the temperature parameter. We then apply Algorithm 1 iteratively, tracking the 2-Wasserstein distance to the original word's embedding and the nearest word in the vocabulary according to the 2-Wasserstein distance.

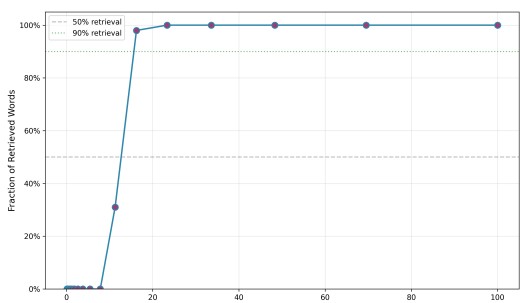

Figure 4: Retrieval success rate vs $\beta$ for Gaussian word embeddings. The plot shows the percentage of correctly retrieved words after 10 iterations of Algorithm 1, with initial perturbations as 2-Wasserstein distance of $1/\sqrt{\beta N}$ from original embeddings. Parameters: $N = 10000$, $d = 50$, spherical covariances.

Figures 4 and 5 illustrate the critical role of the temperature parameter $\beta$ in retrieval convergence for real-world Gaussian word embeddings. At $\beta = 10$ (Figure 5a), the dynamics fail to converge, exhibiting high variance across word samples, whereas at $\beta = 50$ (Figure 5b), the system converges rapidly in a single step with minimal variance. Figure 4 further reveals a sharp phase transition in retrieval success: for $\beta < 10$, retrieval is almost entirely unsuccessful, while around $\beta \approx 15$, the success rate rapidly increases from 0% to 100%, achieving perfect retrieval for $\beta > 30$. This behavior confirms our theoretical prediction that sufficiently large $\beta$ is necessary to create sharp energy basins, satisfy the separation condition in Assumption 1, and ensure contractivity of the $\Phi$ operator. We provide additional results in Section 7.2.4.

## 5 CONCLUSION

In this work, we extended dense associative memories from Euclidean space to the Bures–Wasserstein manifold of Gaussian measures. We proposed a Wasserstein-energy-based memory, derived explicit retrieval maps, and established theoretical guarantees including high-probability retrieval bounds, and exponential storage capacity. Empirically, our Gaussian DAM achieves robust retrieval under perturbations, demonstrating the utility of transport-based aggregation. Conceptually, this framework enables principled reasoning over distributions rather than point estimates, bridging classical associative memories with modern distributional representations. Future directions include extending to broader distribution families (in particular point-cloud data represented as empirical measures), developing particle-based retrieval algorithms, and exploring applications in generative modeling and probabilistic reasoning.

## 6 REPRODUCIBILITY STATEMENT

The Appendix contains all theoretical results, and the supplementary material provides code to reproduce our experiments. LLM was used only to polish the writing.

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

# 7 APPENDIX

## 7.1 RELATED WORK

The study of associative memory begins with Hopfield's seminal model (Hopfield, 1982b) which framed memory recall as gradient descent on a quadratic energy landscape . While conceptually foundational, the quadratic Hopfield energy yields only linear storage capacity in the ambient dimension and suffers from spurious attractors at scale. Recent work revived and substantially extended this line of research by introducing highly nonlinear energy functions that dramatically increase capacity. In particular (Krotov & Hopfield, 2016) proposed log-sum-exp style energies and showed that dense associative memories (DAMs) can realize exponentially many stable patterns relative to dimension. Subsequent developments formalized the exponential capacity rigorously and established connections between modern attention/associative recall mechanisms and Hopfield-style energy landscapes (see, for e.g., Demircigil et al. (2017); Ramsauer et al. (2020); Lucibello & Mézard (2024)), showing both practical and theoretical equivalences between attention-like updates and energy-based recall. We also refer the interested reader to recent works (Krotov & Hopfield, 2018; Krotov, 2021; dos Santos et al., 2024; Hoover et al., 2024b; Hu et al., 2023; Hoover et al., 2024a; Wu et al., 2024; Hoover et al., 2025), including the survey by Krotov et al. (2025) for the state-of-the-art on modern associative memories.

Optimal transport, Wasserstein geometry and barycenters. Optimal transport has emerged as a central tool to compare and interpolate probability measures; the Wasserstein-2 metric in particular induces a rich geometric structure that is especially well behaved on Gaussian families (the so-called Bures–Wasserstein geometry). The mathematical theory of Wasserstein barycenters and their computation was significantly advanced by Agueh and Carlier (Agueh & Carlier, 2011), and efficient numerical algorithms, including entropic regularization approaches, have been developed by Cuturi and Doucet (Cuturi & Doucet, 2014) and others. The computational and theoretical foundations of optimal transport are now well-summarized in recent treatments (Peyré & Cuturi, 2019). Our work leverages these results: stationary points of our distributional log-sum-exp energy are self-consistent barycenters in Wasserstein space, and we exploit closed-form formulas and fixed-point iterations available for Gaussian barycenters to derive concrete retrieval dynamics and guarantees.

Very recently, the generative modeling community has increasingly focused on models and architectures that operate over probability distributions. Rectified Point Flow learns continuous velocity fields for point-cloud registration and assembly (Sun et al., 2025), Wasserstein Flow Matching generalizes flow matching to families of distributions via optimal transport geometry (Haviv et al., 2025), and Bonet et al. (2025) propose flowing measures for distributional generation tasks. These approaches emphasize generative modeling or alignment, whereas our work develops a dense associative memory over probability measures with rigorous capacity and retrieval guarantees, thereby complementing flow-based paradigms. Our approach can be seen as an energy-based generative mechanism in the space of probability measures: fixed points of our Wasserstein log-sum-exp energy yield full probability laws that serve as generative attractors. This perspective unifies associative memory and generative modeling, and suggests novel ways to incorporate memory into uncertainty-aware generative pipelines.

## 7.2 ADDITIONAL EXPERIMENTAL RESULTS

### 7.2.1 SAMPLING EIGENVALUES

First, we present Algorithm 2, which is a minor modification of Algorithm 3 and makes sampling of eigenvalues computationally faster.

### 7.2.2 ADDITIONAL NUMERICAL EXPERIMENTS WITH COMMUTING COVARIANCE

Figure 6 demonstrates the robustness of our retrieval dynamics across different problem scales. While maintaining the same simulation protocol as Figure 3 and parameter settings ($\beta \in \{1, 0.1\}$, $r \in \{1/\sqrt{\beta N}, 100/\sqrt{\beta N}\}$), we test with $N = 5000$ Gaussian measures in dimension $d = 25$, compared to $N = 10000$, $d = 50$ in Figure 3. The consistent convergence behavior across different scales validates that our theoretical results hold for varying dimensions and measure counts.

---

**Algorithm 2** Sampling from commuting Gaussian measures on Wasserstein sphere

---

**Require:** $R > 0$ (sphere radius), $N \in \mathbb{N}$ (number of samples), $\lambda_{\min}, \lambda_{\max} > 0$ (eigenvalue bounds with $\lambda_{\min} \leq \lambda_{\max}$), $d \in \mathbb{N}$ (dimension)

**Ensure:** $\{X_1, \ldots, X_N\}$ where $X_i \sim \mathcal{N}(\mu_i, \Sigma_i)$ on $\mathcal{S}_R$

1: **Initialize:** Fix an orthogonal matrix $U \in \mathbb{R}^{d \times d}$

2: Set target sum $S = \frac{R^2}{2} = \frac{d(\lambda_{\min} + \lambda_{\max})}{2}$

3: **for** $i = 1$ to $N$ **do**

4:     **repeat**

5:         Sample $\lambda_1^{(i)}, \ldots, \lambda_{d-1}^{(i)} \sim \text{Uniform}[\lambda_{\min}, \lambda_{\max}]$

6:         Compute $\lambda_d^{(i)} = S - \sum_{k=1}^{d-1} \lambda_k^{(i)}$

7:     **until** $\lambda_d^{(i)} \in [\lambda_{\min}, \lambda_{\max}]$

8:     Randomly permute $(\lambda_1^{(i)}, \ldots, \lambda_d^{(i)})$ to avoid bias

9:     Construct covariance matrix: $\Sigma_i \leftarrow U \cdot \text{diag}(\lambda_1^{(i)}, \ldots, \lambda_d^{(i)}) \cdot U^T$

10:     Sample mean vector: $\mu_i \sim \text{Uniform}\left( \left\{ \mu \in \mathbb{R}^d : \|\mu\|_2 = \frac{R}{\sqrt{2}} \right\} \right)$

11:     Set $X_i \leftarrow \mathcal{N}(\mu_i, \Sigma_i)$

12: **end for**

13: **return** $\{X_1, \ldots, X_N\}$

---

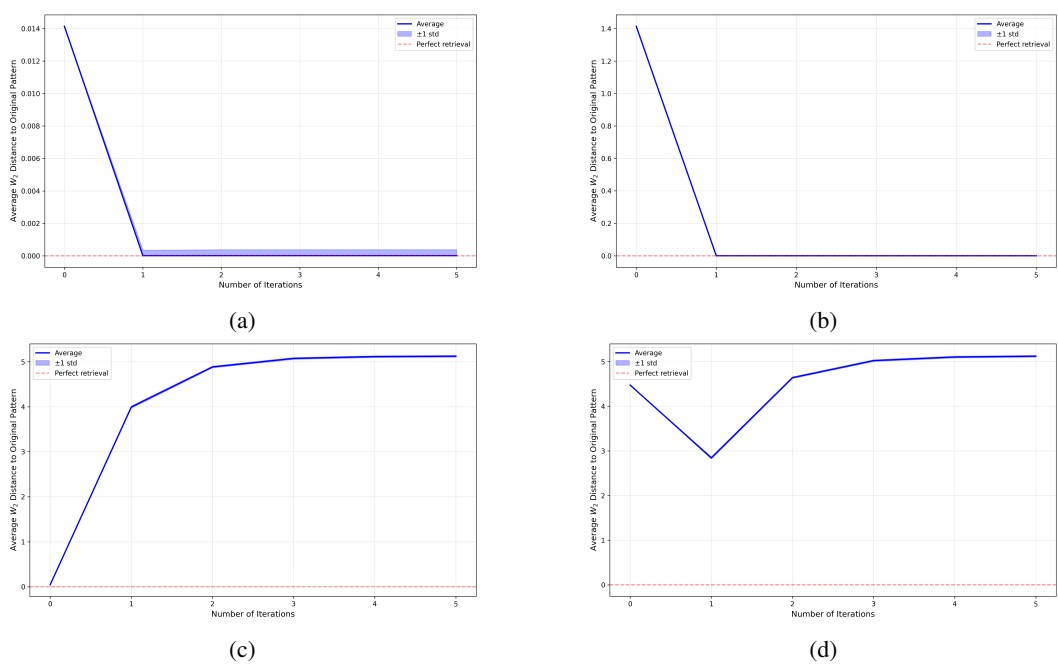

Figure 6: Convergence of retrieval dynamics (Algorithm 1) for perturbed Gaussian measures. Average 2-Wasserstein distance to original measures over iterations for different values of temperature parameter $\beta$ and perturbation radius $r$. (a) $\beta = 1, r = 1/\sqrt{\beta N}$, (b) $\beta = 1, r = 100/\sqrt{\beta N}$, (c) $\beta = 0.1, r = 1/\sqrt{\beta N}$, (d) $\beta = 0.1, r = 100/\sqrt{\beta N}$. Averages computed over 3750 perturbed measures from $N = 5000$ sampled Gaussians in dimension 25 with eigen-value bounds $\lambda_{\min} = 1, \lambda_{\max} = 1.1$. The dotted red horizontal line indicates the convergence threshold at 2-Wasserstein distance of $10^{-6}$.

The one step convergence observed for $\beta = 1$ in Figures 3 and 6 makes the visualization of contours along retrieval dynamics unnecessary as the retrieved and original Gaussian measures essentially overlap. In contrast, the case $\beta = 0.1$ in Figures 3 and 6 exhibits different behavior, with the retrieval dynamics failing to converge to the original pattern. Figure 7 provides a detailed visualization, showing the evolution of one perturbed Gaussian measure through five iterations of Algorithm 1.

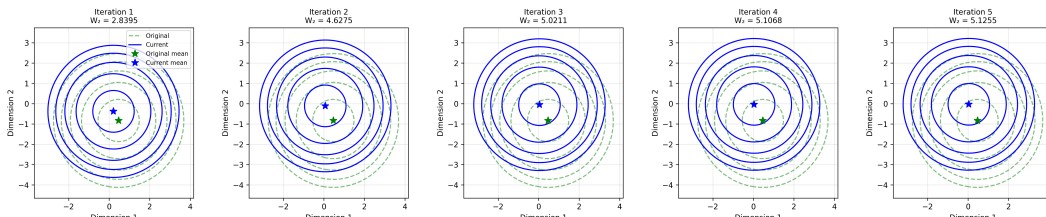

Figure 7: Evolution of a perturbed Gaussian measure over 5 iterations of Algorithm 1 (first 2 dimensions). Green dashed contours represent the original Gaussian measure, while blue solid contours show the current state after each iteration. The Wasserstein distance $W_2$ to the original pattern is displayed above each panel. Stars indicate the mean vectors (green: original, blue: current). The parameter values are $N = 5000, d = 25, \beta = 0.1, \lambda_{\min} = 1, \lambda_{\max} = 1.1$.

The plot reveals that the retrieved distribution diverges from the original Gaussian distribution both in mean and covariance structure.

### 7.2.3 NON-COMMUTING COVARIANCE SIMULATION

To evaluate the retrieval dynamics of Algorithm 1 in the general non-commuting covariance matrices case, we sampled $N = 1000$ Gaussian distributions $X_i = \mathcal{N}(\mu_i, \Sigma_i)$, for $i \in [N]$, from a Wasserstein sphere of radius $R = \sqrt{2d}$ in dimension $d = 10$. Following the approach used for commuting covariances, we allocated the total Wasserstein budget $R^2 = 2d$ equally between the mean and covariance components, setting $\|\mu_i\|^2 = R^2/2 = \text{tr}(\Sigma_i)$ for all $i \in [N]$.

For each Gaussian measure $X_i$, the mean vector was sampled uniformly from the sphere of radius $R/\sqrt{2}$. The covariance matrix was constructed by first generating a matrix $W \in \mathbb{R}^{R^2/2 \times R^2/2}$ with i.i.d. standard Gaussian entries, forming the initial positive semi-definite matrix $WW^T$. To ensure numerical stability in subsequent computations, we added a regularization terms $0.01I$ to $WW^T$. The resultant matrix was then scaled by an appropriate factor to achieve a trace of $R^2/2$. This sampling procedure generates diverse eigenvalues and eigenvectors without imposing a commuting covariance structure.

To test the retrieval dynamics, we randomly selected 750 Gaussian measures (75% of the total) and perturbed each by a Wasserstein distance of $r = 1/\sqrt{\beta N}$. The perturbation was implemented by a split-budget approach, allocating $r^2/2$ to the perturbation of the mean and $r^2/2$ to the covariance perturbation. For the covariance perturbation, we generated a random positive semi-definite perturbation direction and used binary search to find the scaling parameter that achieves the target covariance perturbation. This approach ensures that each perturbed Gaussian measures lies at a 2-Wasserstein distance of $r$ from the original Gaussian measure.

Figure 8 demonstrates the convergence behavior with the above experimental setup. Panels (a), (b) show convergence in one step at $\beta = 1$ similar to the commuting covariance matrices case. However, this convergence has higher variance than in the commuting case and reaches a threshold level of $10^{-3}$ rather than the $10^{-6}$ achieved in the commuting case. Panels (c), (d) show non-convergence at $\beta = 0.1$, consistent with our earlier findings for commuting covariance matrices.

### 7.2.4 ADDITIONAL REAL-DATA EXPERIMENTS

Figure 9 demonstrates the retrieval dynamics of Algorithm 1 of our distributional associative memory on Gaussian word embeddings learned from the Text8 corpus. The temperature parameter $\beta$ critically determines retrieval success: at $\beta = 1$, the dynamics fail to converge to their original embeddings; at $\beta = 10$, the dynamics preserve the word for 1-3 iterations with the word dynamics diverging from their original in subsequent iterations; and at $\beta = 50$, retrieval achieves convergence to the original word in a single iteration. The bottom panels reveal the word-level evolution during retrieval, where perturbed embeddings initially map to the same word and the dynamics of Algorithm 1 preserve the original word for sufficiently large $\beta$. This behavior aligns with our theoretical predictions, where higher $\beta$ values create sharper energy basins that facilitate more robust retrieval, while low $\beta$ values result in overly smooth energy landscapes that prevent proper pattern separation.

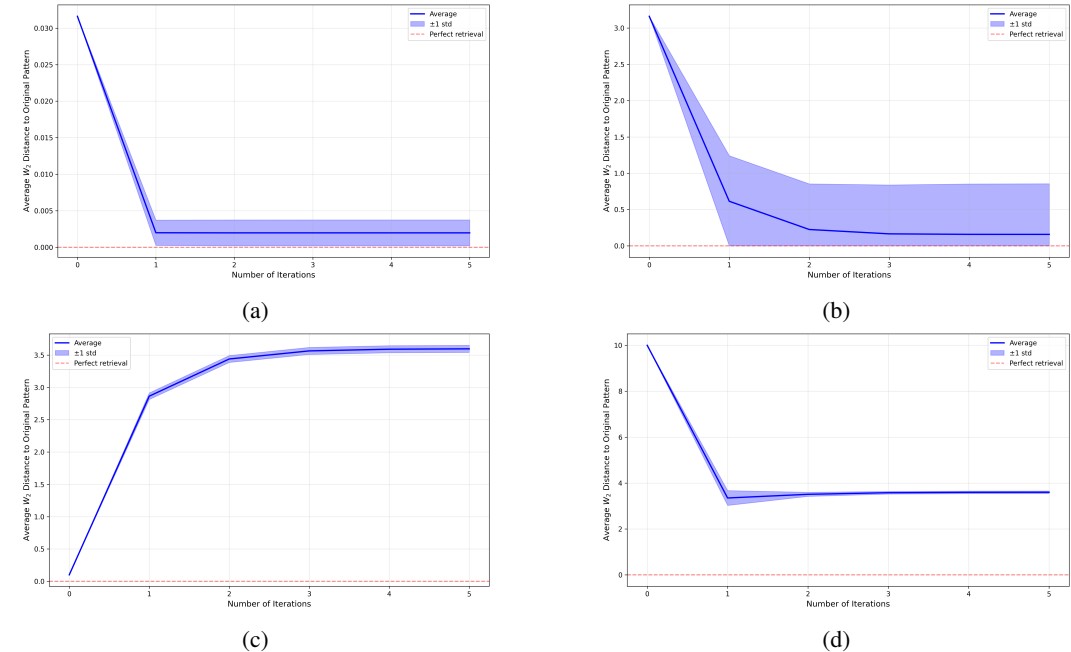

Figure 8: Convergence of retrieval dynamics (Algorithm 1) for perturbed Gaussian measures. Average 2-Wasserstein distance to original measures over iterations for different values of temperature parameter $\beta$ and perturbation radius $r$. (a) $\beta = 1, r = 1/\sqrt{\beta N}$, (b) $\beta = 1, r = 100/\sqrt{\beta N}$, (c) $\beta = 0.1, r = 1/\sqrt{\beta N}$, (d) $\beta = 0.1, r = 100/\sqrt{\beta N}$. Averages computed over 750 perturbed measures from $N = 1000$ sampled Gaussians in dimension 10. The dotted red horizontal line indicates the convergence threshold at 2-Wasserstein distance of $10^{-6}$.

### 7.3 PRELIMINARY RESULTS

In this section, we establish the following three fundamental results which are crucial to prove our results on storage capacity and retrieval rates:

1. The operator $\Phi$ defined by weighted transport maps in equation 5 preserves Gaussian structure (Lemma 2).
2. For sufficiently separated patterns, $\Phi$ maps Wasserstein balls around patterns to themselves (Lemma 3).
3. Within these balls, the operator $\Phi$ is a contraction, guaranteeing convergence to unique fixed points (Lemma 5).

**Lemma 1.** *Let $X_1 = \mathcal{N}(\mu_1, \Sigma_1)$ and $X_2 = \mathcal{N}(\mu_2, \Sigma_2)$. Then*

$$W_2^2(X_1, X_2) = \|\mu_1 - \mu_2\|^2 + \mathsf{tr}(\Sigma_1 + \Sigma_2 - 2(\Sigma_1^{1/2}\Sigma_2\Sigma_1^{1/2})^{1/2}).$$

*Proof of Lemma 1.* This is Lambert et al. (2022)[Equation 5]. □

**Lemma 2.** *Let $X_i = \mathcal{N}(\mu_i, \Sigma_i)$ for $i = 1, 2, \ldots, N$ be the stored patterns. If $\xi = \mathcal{N}(m, \Sigma_0)$, then*

$$\Phi(\xi) = \mathcal{N}(m', \tilde{A}\Sigma_0\tilde{A}^T),$$

*where $m' = \sum_{i=1}^{N} w_i(\xi)\mu_i$ and $\tilde{A} := \sum_{i=1}^{N} w_i(\xi)A_i$.*

*Proof of Lemma 2.* By Asuka (2011)[Lemma 2.3], we have that the optimal transport map from $\xi$ to $X_i$ is

$$T_i(x) = \mu_i + A_i(x - m),$$

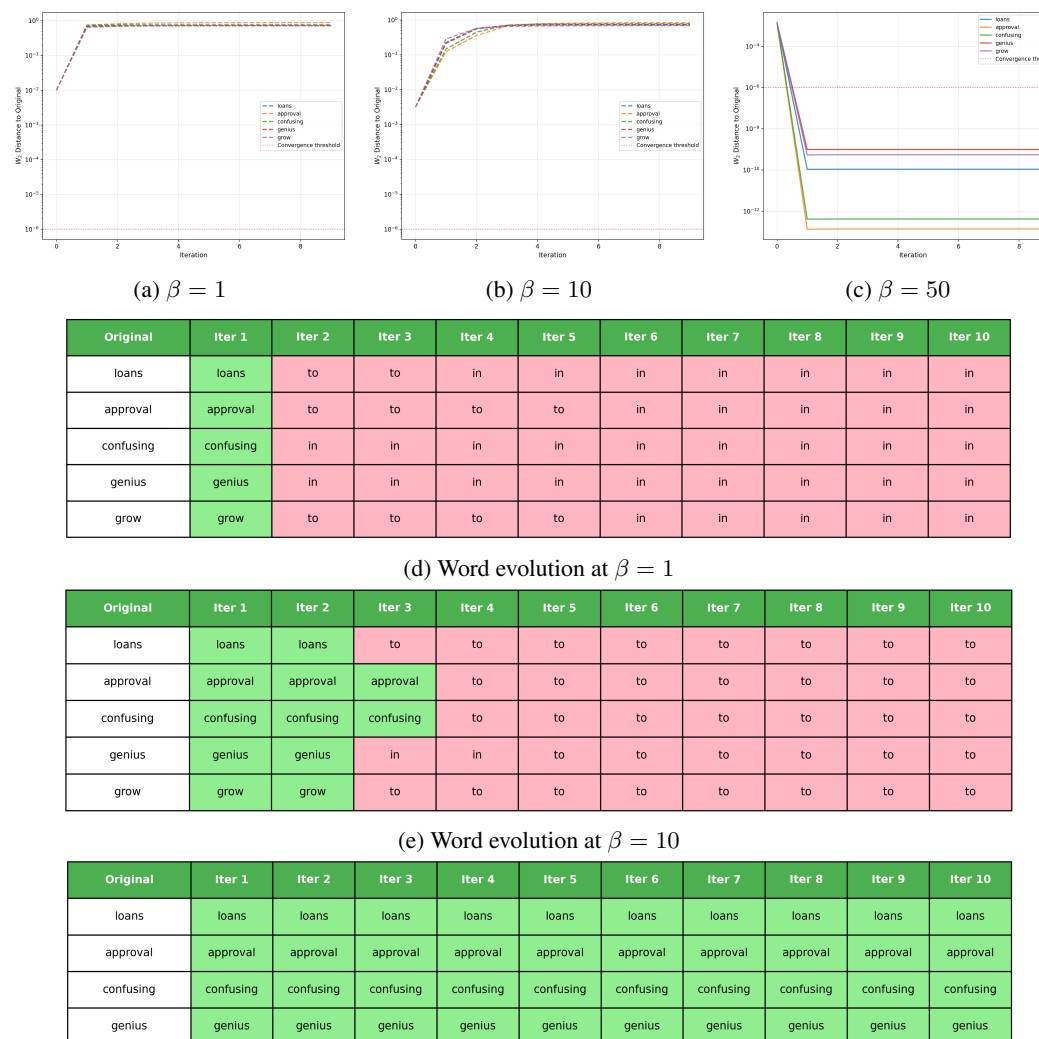

(a) $\beta = 1$      (b) $\beta = 10$      (c) $\beta = 50$

| Original | Iter 1 | Iter 2 | Iter 3 | Iter 4 | Iter 5 | Iter 6 | Iter 7 | Iter 8 | Iter 9 | Iter 10 |
|---|---|---|---|---|---|---|---|---|---|---|
| loans | loans | to | to | in | in | in | in | in | in | in |
| approval | approval | to | to | to | to | in | in | in | in | in |
| confusing | confusing | in | in | in | in | in | in | in | in | in |
| genius | genius | in | in | in | in | in | in | in | in | in |
| grow | grow | to | to | to | to | in | in | in | in | in |

(d) Word evolution at $\beta = 1$

| Original | Iter 1 | Iter 2 | Iter 3 | Iter 4 | Iter 5 | Iter 6 | Iter 7 | Iter 8 | Iter 9 | Iter 10 |
|---|---|---|---|---|---|---|---|---|---|---|
| loans | loans | loans | to | to | to | to | to | to | to | to |
| approval | approval | approval | approval | to | to | to | to | to | to | to |
| confusing | confusing | confusing | confusing | to | to | to | to | to | to | to |
| genius | genius | genius | in | in | to | to | to | to | to | to |
| grow | grow | grow | to | to | to | to | to | to | to | to |

(e) Word evolution at $\beta = 10$

| Original | Iter 1 | Iter 2 | Iter 3 | Iter 4 | Iter 5 | Iter 6 | Iter 7 | Iter 8 | Iter 9 | Iter 10 |
|---|---|---|---|---|---|---|---|---|---|---|
| loans | loans | loans | loans | loans | loans | loans | loans | loans | loans | loans |
| approval | approval | approval | approval | approval | approval | approval | approval | approval | approval | approval |
| confusing | confusing | confusing | confusing | confusing | confusing | confusing | confusing | confusing | confusing | confusing |
| genius | genius | genius | genius | genius | genius | genius | genius | genius | genius | genius |
| grow | grow | grow | grow | grow | grow | grow | grow | grow | grow | grow |

(f) Word evolution at $\beta = 50$

Figure 9: Retrieval dynamics of perturbed Gaussian word embeddings for different temperature parameters. **Top row:** Average 2-Wasserstein distance to original embeddings over iterations. **Bottom rows:** Evolution of retrieved words, showing convergence patterns. Initial perturbation achieves $W_2$ distance of $1/\sqrt{\beta N}$ from original embeddings. Results are for 5 randomly selected words from $N = 10000$ vocabulary.

where $A_i = \Sigma_i^{1/2}(\Sigma_i^{1/2}\Sigma_0\Sigma_i^{1/2})^{-1/2}\Sigma_i^{1/2}$. Therefore, the weighted sum of transport maps is:

$$\sum_{i=1}^{N} w_i(\xi)T_i(x) = \sum_{i=1}^{N} w_i(\xi)\left(\mu_i + A_i(x - m)\right)$$
$$= m' + \tilde{A}(x - m)\,,$$

where $\tilde{A} := \sum_{i=1}^{N} w_i(\xi)A_i$ and $m' := \sum_{i=1}^{N} w_i(\xi)\mu_i$. Thus, the weight sum of transport maps is an affine map. From the definition of operator $\Phi$ in equation 5, we have

$$\Phi(\xi) = S_{\#}\xi\,,$$

where $S = \sum_{i=1}^{N} w_i(\xi) T_i$. If $X \sim \xi = \mathcal{N}(m, \Sigma_0)$, then $S(X) = m' + \tilde{A}(X - m)$. Hence, $\mathbb{E}[S(X)] = m'$ and $\text{Var}(S(X)) = \tilde{A}\text{Cov}(X)\tilde{A}^T = \tilde{A}\Sigma_0\tilde{A}^T$ and

$$\Phi(\xi) = \mathcal{N}(m', \tilde{A}\Sigma_0\tilde{A}^T),$$

where $m' = \sum_{i=1}^{N} w_i(\xi)\mu_i$ and $\tilde{A} := \sum_{i=1}^{N} w_i(\xi) A_i$.

$\square$

The idea is to apply Banach's fixed point theorem to prove the existence of a unique fixed point around each pattern. To that end, first we prove that $\Phi$ is a self-map in a neighborhood around each pattern.

**Lemma 3.** *Let $X_i = \mathcal{N}(\mu_i, \Sigma_i)$ for $i = 1, 2, \ldots N$ be the given patterns, where $\mu_i \in \mathbb{R}^d$ and $\Sigma_i \succ 0$. Define the Wasserstein ball $B_i = \{\nu \in \mathcal{P}_2(\mathbb{R}^d) : W_2(X_i, \nu) \leq r\}$ where $r = \frac{1}{\sqrt{\beta N}}$. Let $\xi = \mathcal{N}(m, \Omega) \in B_i$ with $\Omega \succ 0$ be the query measure. Assume that all the covariance matrices commute pairwise: $\Sigma_i \Sigma_j = \Sigma_j \Sigma_i$ for all $i, j$ and $\Sigma_i \Omega = \Omega \Sigma_i$ for all $i$. Let the eigenvalues of all $X_i$ and $\xi$ lie in a bounded interval $[\lambda_{\min}, \lambda_{\max}]$. Define the operator $\Phi : \mathcal{P}_2(\mathbb{R}^d) \to \mathcal{P}_2(\mathbb{R}^d)$ as:*

$$\Phi(\xi) := \left( \sum_{j=1}^{N} w_j(\xi) T_j \right)_{\#} \xi,$$

*where $T_j$ is the optimal transport map from $\xi$ to $X_j$ and the weights are*

$$w_j(\xi) = \frac{\exp(-\beta W_2^2(X_j, \xi))}{\sum_{k=1}^{N} \exp(-\beta W_2^2(X_k, \xi))}.$$

*Define*

*1. $M_W := \max_i W_2(X_i, \delta_0)$ where $\delta_0$ is the delta measure at the origin*

*2. $\Delta_i := \min_{j \neq i}(-\log\langle X_i, X_j\rangle_{L^2})$*

*If the separation condition $\Delta_i$ satisfies*

$$\Delta_i \geq \frac{d}{2}\log(4\pi\lambda_{\max}) + \frac{1}{\beta\lambda_{\min}}\log\left(N^3\beta(4M_W^2 + 2d(\lambda_{\max} + \lambda_{\min}))\right),$$

*and $\left(4M_W^2 + 2d(\lambda_{\max} + \lambda_{\min})\right)N^3\beta > e^2$, then*

$$\Phi(\xi) \in B_i.$$

*Proof of Lemma 3.* First, since $\Sigma_i, \Sigma_j$ commute for all $i, j$ and $\Sigma_i$ commutes with $\Omega$ for all $i$, we can diagonalize $\Sigma_i, \Omega$ in a common eigenbasis. In particular, there exists an orthogonal matrix $U$ such that

$$\Sigma_i = U\text{diag}(\lambda_{i,1}, \ldots, \lambda_{i,d})U^T, \quad \Omega = U\text{diag}(\omega_1, \ldots, \omega_d)U^T,$$

for all $i \in [N]$, where $\text{diag}(\lambda_{i,1}, \ldots, \lambda_{i,d})$ and $\text{diag}(\omega_1, \ldots, \omega_d)$ are diagonal matrices containing eigenvalues of $\Sigma_i, \Omega$ respectively.

Next, we relate the $L^2$ inner product between $X_1, X_2$ to the 2-Wasserstein distance between them. By definition of the $L^2$ inner product and since $\Sigma_i, \Sigma_j$ share a common eigenbasis $U$, we have

$$-\log\langle X_i, X_j\rangle_{L^2} = \frac{d}{2}\log(2\pi) + \frac{1}{2}\log|\Sigma_i + \Sigma_j| + \frac{1}{2}(\mu_i - \mu_j)^T(\Sigma_1 + \Sigma_2)^{-1}(\mu_i - \mu_j)$$

$$= \frac{d}{2}\log(2\pi) + \frac{1}{2}\sum_{k=1}^{d}\log(\lambda_{i,k} + \lambda_{j,k}) + \frac{1}{2}(\mu_i - \mu_j)^T\text{diag}\left(\frac{1}{\lambda_{i,1} + \lambda_{j,1}}, \ldots, \frac{1}{\lambda_{i,d} + \lambda_{j,d}}\right)(\mu_i - \mu_j)$$

$$= \frac{d}{2}\log(2\pi) + \frac{1}{2}\sum_{k=1}^{d}\log(\lambda_{i,k} + \lambda_{j,k}) + \frac{1}{2}\sum_{k=1}^{d}\frac{[U^T(\mu_i - \mu_j)]_k^2}{\lambda_{i,k} + \lambda_{j,k}}.$$

Since $\log(\lambda_{i,k} + \lambda_{j,k}) \leq \log(2\lambda_{\max})$, and $-\log\langle X_i, X_j \rangle_{L^2} \geq \Delta_i$, we obtain

$$\sum_{k=1}^{d} \frac{[U^T(\mu_i - \mu_j)]_k^2}{\lambda_{i,k} + \lambda_{j,k}} \geq 2\left(\Delta_i - \frac{d}{2}\log(2\pi) - \frac{d}{2}\log(2\lambda_{\max})\right).$$

Next, since $\lambda_{i,k} + \lambda_{j,k} \geq 2\lambda_{\min}$, we get

$$\|\mu_i - \mu_j\|^2 = \sum_{k=1}^{d} [U^T(\mu_i - \mu_j)]_k^2 \geq 4\lambda_{\min}\left(\Delta_i - \frac{d}{2}\log(4\pi\lambda_{\max})\right) =: D. \tag{6}$$

Next, since the weight $w_i$ is given by

$$w_i(\xi) = \frac{\exp\left(-\beta W_2^2(X_i, \xi)\right)}{\sum_{j=1}^{N} \exp\left(-\beta W_2^2(X_j, \xi)\right)} = \frac{1}{1 + \sum_{j \neq i} \exp\left(-\beta[W_2^2(X_j, \xi) - W_2^2(X_i, \xi)]\right)}, \tag{7}$$

we first show that $w_i(\xi)$ is large and $w_j(\xi)$, for all $j \neq i$, is small by showing that $W_2^2(X_j, \xi) - W_2^2(X_i, \xi)$ is large for all $j \neq i$. By the triangle inequality, the assumption $\xi \in B_i$, and from equation 6, we get

$$W_2(X_j, \xi) \geq W_2(X_j, X_i) - W_2(X_i, \xi) \geq W_2(X_j, X_i) - r \geq \sqrt{D} - r.$$

By squaring both the sides of the above inequality and subtracting $W_2^2(X_i, \xi)$, we get:

$$W_2^2(X_j, \xi) - W_2^2(X_i, \xi) \geq D - 2\sqrt{D}r.$$

By plugging the above inequality into equation 7, we obtain:

$$w_i(\xi) \geq \frac{1}{1 + (N-1)\exp(-\beta(D - 2\sqrt{D}r))}.$$

By defining $\varepsilon := (N-1)\exp(-\beta(D - 2\sqrt{D}r))$ and using $\frac{1}{1+\varepsilon} \geq 1 - \varepsilon$ for all $\varepsilon > 0$, we get $w_i(\xi) \geq 1 - \varepsilon$.

Next, we find an upper bound on $W_2(\Phi(\xi), X_i)$. By Lemma 1 and Lemma 2, we have

$$W_2^2(\Phi(\xi), X_i) = \|m' - \mu_i\|^2 + \mathrm{tr}(\tilde{A}\Omega\tilde{A}^T + \Sigma_i - 2((\tilde{A}\Omega\tilde{A}^T)^{1/2}\Sigma_i(\tilde{A}\Omega\tilde{A}^T)^{1/2}), \tag{8}$$

where $m' = \sum_{i=1}^{N} w_i(\xi)\mu_i$ and $\tilde{A} := \sum_{i=1}^{N} w_i(\xi)A_i$. We refer to the above first term as the *mean error* and the second term as the *covariance error*.

First, we bound the mean error. Using $\|\mu_k\| \leq W_2(X_k, \delta_0) \leq M_W$ for all $k$, $w_i(\xi) = 1 - \sum_{j \neq i} w_j(\xi)$, and Jensen's inequality, we get:

$$\|m' - \mu_i\|^2 = \|(w_i(\xi) - 1)\mu_i + \sum_{j \neq i} w_j(\xi)\mu_j\|^2 = \|\sum_{j \neq i} w_j(\xi)(\mu_j - \mu_i)\|^2 \leq \sum_{j \neq i} w_j(\xi)\|\mu_j - \mu_i\|^2 \leq 4\varepsilon M_W^2. \tag{9}$$

Next, we bound the covariance error. To simplify notation in the next steps, we define:

$$\mathsf{CovErr}(\Sigma_1, \Sigma_2) := \mathrm{tr}(\Sigma_1 + \Sigma_2 - 2(\Sigma_1^{1/2}\Sigma_2\Sigma_1^{1/2})^{1/2}).$$

Using this notation, the covariance error in equation 8 is $\mathsf{CovErr}(\tilde{A}\Omega\tilde{A}^T, \Sigma_i)$. By definition,

$$A_i = \Sigma_i^{1/2}(\Sigma_i^{1/2}\Omega\Sigma_i^{1/2})^{-1/2}\Sigma_i^{1/2},$$

and since $\Sigma_i, \Omega$ commute for all $i \in [N]$, $\Sigma_i = U\mathsf{diag}(\lambda_{i,1}, \ldots, \lambda_{i,d})U^T$, and $\Omega = U\mathsf{diag}(\omega_1, \ldots, \omega_d)U^T$, we have

$$A_i = U\mathsf{diag}\left(\sqrt{\frac{\lambda_{i,j}}{\omega_j}}\right)_{j=1}^{d} U^T.$$

By plugging the above into the definition of $\tilde{A}$, we obtain

$$\tilde{A} = \sum_{i=1}^{N} w_i(\xi) A_i = U \mathsf{diag}\left(\sum_{k=1}^{N} w_k(\xi)\sqrt{\frac{\lambda_{k,j}}{\omega_j}}\right)_{j=1}^{d} U^T .$$

Therefore,

$$\tilde{A}\Omega\tilde{A}^T = U\mathsf{diag}\left(\left(\sum_{k=1}^{N} w_k(\xi)\sqrt{\lambda_{k,j}}\right)^2\right)_{j=1}^{d} U^T .$$

Since $\tilde{A}\Omega\tilde{A}^T$ is diagonal in the basis $U$, we have

$$(\tilde{A}\Omega\tilde{A}^T)^{1/2} = U\mathsf{diag}\left(\left(\sum_{k=1}^{N} w_k(\xi)\sqrt{\lambda_{k,j}}\right)\right)_{j=1}^{d} U^T . \qquad (10)$$

From equation 10 and $\Sigma_i = U\mathsf{diag}(\lambda_{i,1},\dots,\lambda_{i,d})U^T$ , we obtain:

$$\left((\tilde{A}\Omega\tilde{A}^T)^{1/2}\Sigma_i(\tilde{A}\Omega\tilde{A}^T)^{1/2}\right)^{1/2} = U\mathsf{diag}\left(\sqrt{\lambda_{i,j}}\sum_{k=1}^{N} w_k(\xi)\sqrt{\lambda_{k,j}}\right)_{j=1}^{d} U^T .$$

Applying the cyclic property of the trace to the above expression yields

$$\mathsf{tr}\left(\left((\tilde{A}\Omega\tilde{A}^T)^{1/2}\Sigma_i(\tilde{A}\Omega\tilde{A}^T)^{1/2}\right)^{1/2}\right) = \mathsf{tr}\left(\mathsf{diag}\left(\sqrt{\lambda_{i,j}}\sum_{k=1}^{N} w_k(\xi)\sqrt{\lambda_{k,j}}\right)_{j=1}^{d}\right) = \sum_{j=1}^{d}\sqrt{\lambda_{i,j}}\sum_{k=1}^{N} w_k(\xi)\sqrt{\lambda_{k,j}} .$$

Next, by plugging the above expression into the definition of $\mathsf{CovErr}(\tilde{A}\Omega\tilde{A}^T, \Sigma_i)$, by using $w_i(\xi) - 1 = -\sum_{k\neq i} w_k(\xi)$, $(a-b)^2 \leq 2(a^2+b^2)$ and $\sum_{k\neq i} w_k(\xi) = 1 - w_i(\xi) \leq \varepsilon$, we obtain

$$\mathsf{CovErr}(\tilde{A}\Omega\tilde{A}^T, \Sigma_i) = \sum_{j=1}^{d}\left(\sum_{k=1}^{N} w_k(\xi)\sqrt{\lambda_{k,j}}\right)^2 + \sum_{j=1}^{d}\lambda_{i,j} - 2\sum_{j=1}^{d}\sqrt{\lambda_{i,j}}\sum_{k=1}^{N} w_k(\xi)\sqrt{\lambda_{k,j}}$$

$$= \sum_{j=1}^{d}\left(\sum_{k=1}^{N} w_k(\xi)\sqrt{\lambda_{k,j}} - \sqrt{\lambda_{i,j}}\right)^2$$

$$= \sum_{j=1}^{d}\left[(w_i(\xi) - 1)\sqrt{\lambda_{i,j}} + \sum_{k\neq i} w_k(\xi)\sqrt{\lambda_{k,j}}\right]^2$$

$$= \sum_{j=1}^{d}\left[-\sum_{k\neq i} w_k(\xi)\sqrt{\lambda_{i,j}} + \sum_{k\neq i} w_k(\xi)\sqrt{\lambda_{k,j}}\right]^2$$

$$= \sum_{j=1}^{d}\left(\sum_{k\neq i} w_k(\xi)(\sqrt{\lambda_{k,j}} - \sqrt{\lambda_{i,j}})\right)^2$$

$$\leq \sum_{j=1}^{d}(\sqrt{\lambda_{\mathsf{max}}} - \sqrt{\lambda_{\mathsf{min}}})^2\left(\sum_{k\neq i} w_k(\xi)\right)^2$$

$$\leq 2d(\lambda_{\mathsf{max}} + \lambda_{\mathsf{min}})\varepsilon^2 . \qquad (11)$$

Therefore, by adding the bound on the mean error in equation 9 and the covariance error in equation 11, we get

$$W_2^2(\Phi(\xi), X_i) \leq \left(4M_W^2 + 2d(\lambda_{\mathsf{max}} + \lambda_{\mathsf{min}})\right)\varepsilon^2 = C\varepsilon^2 ,$$

where $C := 4M_W^2 + 2d(\lambda_{\max} + \lambda_{\min})$. The next steps show that $C\varepsilon^2 < r^2$, which would finish the proof. Since

$$\Delta_i \geq \frac{d}{2}\log(4\pi\lambda_{\max}) + \frac{1}{\beta\lambda_{\min}}\log\left(N^3\beta(4M_W^2 + 2d(\lambda_{\max} + \lambda_{\min}))\right),$$

we have by definition of $D$,

$$D = 4\lambda_{\min}\left(\Delta_i - \frac{d}{2}\log(4\pi\lambda_{\max})\right) \geq \frac{4}{\beta}\log\left(CN^3\beta\right) = 8L, \tag{12}$$

where $L := \frac{1}{2\beta}\log(CN^3\beta)$. Next, by definition of $\varepsilon$, we have $\varepsilon < N^2\exp(-2\beta(D - 2\sqrt{D}r))$. Substituting $r = \frac{1}{\sqrt{\beta N}}$, we get $C\varepsilon^2 < CN^2\exp(-2\beta D + 4\sqrt{\beta D/N})$. To show that $C\varepsilon^2 < r^2$, it suffices to prove

$$CN^3\beta\exp(-2\beta D)\exp(4\sqrt{\beta D/N}) < 1.$$

Taking logarithms, this is equivalent to $\log(CN^3\beta) - 2\beta D + 4\sqrt{\beta D/N} < 0$. From the definition of $L$ and dividing by $2\beta$, this condition is equivalent to showing $L - D + 2\frac{\sqrt{D}}{\sqrt{\beta N}} < 0$. Define $f(D) := L - D + 2\frac{\sqrt{D}}{\sqrt{\beta N}}$. We will show that $f(D) < 0$, which will finish the proof. Note that $f'(D) < 0$ if $D > \frac{1}{\beta N}$. From equation 12 and the assumption $CN^3\beta > e^2$, we have

$$D \geq 8L = \frac{4}{\beta}\log(CN^3\beta) \geq \frac{8}{\beta} > \frac{8}{\beta N}.$$

Therefore, $f(D) < f\left(\frac{8}{\beta N}\right)$ for $D > \frac{8}{\beta N}$. Again using the assumption $CN^3\beta > e^2$ in the definition of $L$, we obtain

$$f\left(\frac{8}{\beta N}\right) < \frac{1}{\beta N} - \frac{8}{\beta N} + 2\frac{2\sqrt{2}}{\beta N} < 0,$$

and hence $f(D) < 0$, which finishes the proof.

$\square$

The following lemma proves that the geodesic interpolation of two Gaussian measures with commuting covariance matrices also commutes with the two covariance matrices. Additionally, the lemma finds all the eigenvalues of the covariance matrix of the geodesic interpolation in terms of the eigenvalues of the two covariance matrices. This result is then used in Lemma 5 to prove that $\Phi$ is a contraction mapping.

**Lemma 4.** *Let $\xi_1 = \mathcal{N}(m_1, \Omega_1)$, $\xi_2 = \mathcal{N}(m_2, \Omega_2)$ be two normal distributions in $\mathbb{R}^d$ with $\Omega_1, \Omega_2 \succ 0$. Suppose $\Omega_1, \Omega_2$ commute. Let $\xi_t = \mathcal{N}(m_t, \Omega_t)$ be the geodesic interpolation in Wasserstein space from $\xi_1$ to $\xi_2$. Then*

   *1. $\Omega_t$ commutes with $\Omega_1, \Omega_2$ for all $t \in [0, 1]$*

   *2. The $i$-th eigenvalue $\omega_i(t)$ of $\Omega_t$ is given by*

$$\omega_i(t) = \left((1-t) + t\sqrt{\omega_{2,i}/\omega_{1,i}}\right)^2 \omega_{1,i},$$

   *where $\omega_{1,i}, \omega_{2,i}$ are the $i$-th eigenvalues of $\Omega_1, \Omega_2$ respectively.*

*Proof of Lemma 4.* By Lemma 2, the optimal transport map from $\xi_1$ to $\xi_2$ has the form $T(x) = m_2 + A(x - m_1)$ with $A = \Omega_2^{1/2}(\Omega_2^{1/2}\Omega_1\Omega_2^{1/2})^{-1/2}\Omega_2^{1/2}$. Since $\Omega_1, \Omega_2$ commute, they can be diagonalized by an orthogonal matrix $U$ such that $\Omega_1 = U\Lambda_1 U^T$, $\Omega_2 = U\Lambda_2 U^T$ where $\Lambda_1, \Lambda_2$ are diagonal matrices.

The commutativity of $\Omega_1, \Omega_2$ implies $\Omega_1, \Omega_2^{1/2}$ also commute and so, $(\Omega_2^{1/2}\Omega_1\Omega_2^{1/2})^{-1/2} = (\Omega_2\Omega_1)^{-1/2}$. Hence, the matrix $A$ simplifies to

$$A = \Omega_2^{1/2}(\Omega_2^{1/2}\Omega_1\Omega_2^{1/2})^{-1/2}\Omega_2^{1/2}$$

$$= U\Lambda_2^{1/2}(\Lambda_2\Lambda_1)^{-1/2}\Lambda_2^{1/2}U^T = U\Lambda_2^{1/2}\Lambda_1^{-1/2}U^T .$$

Next, note that the geodesic interpolation $\xi_t = ((1-t)I + tT)_\#\xi_1$ is the pushforward of $\xi_1$ via the map $S_t(x) = ((1-t)I + tA)x + (tm_2 - tAm_1)$. Since the pushforward of a Gaussian $\mathcal{N}(m,\Omega)$ through an affine map $x \to Cx + d$ is $\mathcal{N}(Cm + d, C\Omega C^T)$, we have $\xi_t = \mathcal{N}(m_t, \Omega_t)$ where

$$m_t = (1-t)m_1 + tm_2$$

$$\Omega_t = ((1-t)I + tA)\Omega_1((1-t)I + tA)^T . \tag{13}$$

Define $B := (1-t)I + tA = (1-t)I + tU\Lambda_2^{1/2}\Lambda_1^{-1/2}U^T$. This means

$$BU = U((1-t)I + t\Lambda_2^{1/2}\Lambda_1^{-1/2}) , (BU)^T = ((1-t)I + t\Lambda_1^{-1/2}\Lambda_2^{1/2})U^T .$$

By plugging the above expressions for $BU, (BU)^T$ into equation 13, we get:

$$\Omega_t = B\Omega_1 B^T = BU\Lambda_1 U^T B^T = U((1-t)I + t\Lambda_2^{1/2}\Lambda_1^{-1/2})\Lambda_1((1-t)I + t\Lambda_1^{-1/2}\Lambda_2^{1/2})U^T .$$

This means $\Omega_t = U\Lambda_t U^T$ where $\Lambda_t := ((1-t)I + t\Lambda_2^{1/2}\Lambda_1^{-1/2})\Lambda_1((1-t)I + t\Lambda_1^{-1/2}\Lambda_2^{1/2})$ and $\Omega_t$ can be diagonalized by the same matrix $U$ that diagonalizes $\Omega_1, \Omega_2$. Therefore, $\Omega_t$ commutes with $\Omega_1, \Omega_2$ for all $t \in [0, 1]$.

Next, by the above definition of $\Lambda_t$, note that

$$[\Lambda_t]_{ii} = ((1-t) + t\sqrt{\omega_{2,i}/\omega_{1,i}})\omega_{1,i}((1-t) + t\sqrt{\omega_{2,i}/\omega_{i,i}}) = ((1-t) + t\tau_i)^2\omega_{1,i} ,$$

where $\omega_{1,i}, \omega_{2,i}$ are the $i$-th eigenvalues of $\Omega_1, \Omega_2$ respectively and $\tau_i := \sqrt{\omega_{2,i}/\omega_{1,i}}$ for $i = 1, 2, \ldots, d$. $\qquad\square$

The next lemma proves that the operator $\Phi$ is a contraction mapping.

**Lemma 5.** *Let $X_i = \mathcal{N}(\mu_i, \Sigma_i)$ for $i = 1, 2, \ldots N$ be the given patterns, where $\mu_i \in \mathbb{R}^d$ and $\Sigma_i \succ 0$. Define the Wasserstein ball $B_i = \{\nu \in \mathcal{P}_2(\mathbb{R}^d) : W_2(X_i, \nu) \leq r\}$ where $r = \frac{1}{\sqrt{\beta N}}$. Let $\xi_k = \mathcal{N}(m_k, \Omega_k) \in B_i$ with $\Omega_k \succ 0$ be two query measures, where $k \in \{1, 2\}$. Assume that all the covariance matrices commute pairwise: $\Sigma_i\Sigma_j = \Sigma_j\Sigma_i$ for all $i, j \in [N]$ and $\Sigma_i\Omega_k = \Omega_k\Sigma_i$ for all $i \in [N], k \in \{1, 2\}$. Let the eigenvalues of all $X_i, \xi_1, \xi_2$ lie in a bounded interval $[\lambda_{\min}, \lambda_{\max}]$. Define the operator $\Phi : \mathcal{P}_2(\mathbb{R}^d) \to \mathcal{P}_2(\mathbb{R}^d)$ as:*

$$\Phi(\xi) := \left(\sum_{j=1}^{N} w_j(\xi)T_j\right)_\# \xi ,$$

*where $T_j$ is the optimal transport map from $\xi$ to $X_j$ and the weights are*

$$w_j(\xi) = \frac{\exp(-\beta W_2^2(X_j, \xi))}{\sum_{k=1}^{N}\exp(-\beta W_2^2(X_k, \xi))} .$$

*Define*

    *1. $M_W := \max_i W_2(X_i, \delta_0)$ where $\delta_0$ is the delta measure at the origin*

    *2. $\Delta_i := \min_{j\neq i}(-\log\langle X_i, X_j\rangle_{L^2})$*

*Assume*

    *1.*

$$\Delta_i \geq \frac{d}{2}\log(4\pi\lambda_{\max}) + \frac{1}{\beta\lambda_{\min}}\log\left(N^3\beta(4M_W^2 + 2d(\lambda_{\max} + \lambda_{\min}))\right) .$$

$$\text{(Assumption: Separation)}$$

    *2.*

$$\beta > \frac{e^2}{(4M_W^2 + 2d(\lambda_{\max} + \lambda_{\min}))\, N^3} .$$

$$\text{(Assumption: Constraint)}$$

*If $N > \max\left\{144\beta M_W^2, \frac{1}{(4M_W^2 + 2d(\lambda_{\max} + \lambda_{\min}))\beta}\right\}$, then*

$$W_2(\Phi(\xi_1), \Phi(\xi_2)) \le \kappa W_2(\xi_1, \xi_2),$$

*where $0 < \kappa = \frac{144\beta M_W^2}{N} < 1$.*

*Proof of Lemma 5.* Since all the covariance matrices commute, there exists an orthogonal matrix $U$ such that

$$\Sigma_i = U\text{diag}(\lambda_{i,1}, \ldots, \lambda_{i,d})U^T, \ \Omega_k = U\text{diag}(\omega_{k,1}, \ldots, \omega_{k,d})U^T,$$

for all $i \in [N]$ and $k \in \{1, 2\}$. By Asuka (2011)[Lemma 2.3], we have that for Gaussian measures $\xi_k = \mathcal{N}(m_k, \Omega_k)$ and $X_j = \mathcal{N}(\mu_j, \Sigma_j)$, the optimal transport map from $\xi_k$ to $X_j$ is

$$T_j^{(k)}(x) = \mu_j + A_j^{(k)}(x - m_k),$$

where $A_j^{(k)} := \Sigma_j^{1/2}(\Sigma_j^{1/2}\Omega_k\Sigma_j^{1/2})^{-1/2}\Sigma_j^{1/2}$. Since $\Sigma_j$ and $\Omega_k$ commute and are diagonal in the basis $U$:

$$(\Sigma_j^{1/2}\Omega_k\Sigma_j^{1/2})^{-1/2} = U\text{diag}\left(\frac{1}{\sqrt{\lambda_{j,1}\omega_{k,1}}}, \ldots, \frac{1}{\sqrt{\lambda_{j,d}\omega_{k,d}}}\right)U^T.$$

By plugging the above into the definition of $A_j^{(k)}$, we get

$$A_j^{(k)} = U\text{diag}\left(\frac{\sqrt{\lambda_{j,1}}}{\sqrt{\omega_{k,1}}}, \ldots, \frac{\sqrt{\lambda_{j,d}}}{\sqrt{\omega_{k,d}}}\right)U^T.$$

By Lemma 2, we have $\Phi(\xi_k) = \mathcal{N}(m_k', \tilde{A}\Omega_k\tilde{A}^T)$, where $m_k' = \sum_{j=1}^{N} w_j(\xi_k)\mu_j$ and $\tilde{A}_k = \sum_{j=1}^{N} w_j(\xi_k)A_j^{(k)}$. This means

$$\tilde{A}_k\Omega_k\tilde{A}_k^T = U\text{diag}\left(\left[\sum_{j=1}^{N} w_j(\xi_k)\sqrt{\lambda_{j,\ell}}\right]^2\right)_{\ell=1}^{d} U^T. \tag{14}$$

Additionally, note that if positive definite matrices $P, Q$ commute, then

$$\text{tr}\left(P + Q - 2(P^{1/2}QP^{1/2})^{1/2}\right) = \text{tr}\left(P + Q - 2(PQ)^{1/2}\right)$$

$$= \sum_{\ell=1}^{d} \lambda_\ell(P) + \lambda_\ell(Q) - 2\sqrt{\lambda_\ell(P)\lambda_\ell(Q)}$$

$$= \sum_{\ell=1}^{d} \left(\sqrt{\lambda_\ell(P)} - \sqrt{\lambda_\ell(Q)}\right)^2. \tag{15}$$

Next, by Lemma 1, we have

$$W_2^2(\Phi(\xi_1), \Phi(\xi_2)) = \|m_1' - m_2'\|^2 + \text{tr}\left(\tilde{A}_1\Omega_1\tilde{A}_1^T + \tilde{A}_2\Omega_2\tilde{A}_2^T - 2\left((\tilde{A}_1\Omega_1\tilde{A}_1^T)^{1/2}\tilde{A}_2\Omega_2\tilde{A}_2^T(\tilde{A}_1\Omega_1\tilde{A}_1^T)^{1/2}\right)^{1/2}\right). \tag{16}$$

The fact that $\Omega_1, \Omega_2$ commute means that $\tilde{A}_1\Omega_1\tilde{A}_1^T$ commutes with $\tilde{A}_2\Omega_2\tilde{A}_2^T$ because $\tilde{A}_1\Omega_1\tilde{A}_1^T, \tilde{A}_2\Omega_2\tilde{A}_2^T$ share the same eigenbasis $U$. By using this fact, along with plugging equation 14 and equation 15 into equation 16, and by the definition of $m_k'$, we obtain:

$$W_2^2(\Phi(\xi_1), \Phi(\xi_2)) = \|m_1' - m_2'\|^2 + \sum_{\ell=1}^{d}\left(\sum_{j=1}^{N} w_j(\xi_1)\sqrt{\lambda_{j,\ell}} - \sum_{j=1}^{N} w_j(\xi_2)\sqrt{\lambda_{j,\ell}}\right)^2$$

$$= \left\|\sum_{j=1}^{N} \Delta w_j \mu_j\right\|^2 + \sum_{\ell=1}^{d}\left(\sum_{j=1}^{N} \Delta w_j \sqrt{\lambda_{j,\ell}}\right)^2, \tag{17}$$

where $\Delta w_j := w_j(\xi_1) - w_j(\xi_2)$. The next steps of the proof obtain a bound on $|\Delta w_j|$ so that we can obtain an upper bound for equation 17. Toward that end, we first obtain bounds on weights $w_k(\xi)$ for all $k \in [N]$.

Since $W_2(X_i, X_j) \geq \|\mu_i - \mu_j\|$, we get from equation 6 that $W_2(X_i, X_j) \geq \sqrt{D}$, where $D := 4\lambda_{\min}\left(\Delta_i - \frac{d}{2}\log(4\pi\lambda_{\max})\right)$. By the triangle inequality, $W_2(X_i, X_j) \geq \sqrt{D}$, and $W_2(X_i, \xi) \leq r$, we have

$$W_2(X_j, \xi) \geq W_2(X_j, X_i) - W_2(X_i, \xi) \geq \sqrt{D} - r\,.$$

This means

$$W_2^2(X_j, \xi) - W_2^2(X_i, \xi) > D - 2\sqrt{D}r\,. \tag{18}$$

First, we obtain a lower bound on the weight $w_i(\xi)$. By the definition of the weights $w_i(\xi)$, using equation 18 and $\frac{1}{1+\alpha} > 1 - \alpha$ for all $\alpha > 0$, and using equation Assumption: Separation, we get

$$
\begin{aligned}
w_i(\xi) &= \frac{\exp(-\beta W_2^2(X_i, \xi))}{\sum_{k=1}^N \exp(-\beta W_2^2(X_k, \xi))} \\
&= \frac{1}{1 + \sum_{k \neq i} \exp(-\beta(W_2^2(X_j, \xi) - W_2^2(X_i, \xi)))} \\
&\geq \frac{1}{1+\varepsilon} \\
&\geq 1 - \varepsilon\,,
\end{aligned}
\tag{19}
$$

where

$$\varepsilon := (N-1)\exp\left(-\beta(D - 2\sqrt{D}r)\right)\,. \tag{20}$$

Now, we obtain an upper bound on the weight $w_j(\xi)$ for $j \neq i$. By definition of $w_j(\xi)$ and again using equation 18, we obtain

$$w_j(\xi) = \frac{\exp(-\beta W_2^2(X_j, \xi))}{\sum_{k=1}^N \exp(-\beta W_2^2(X_k, \xi))} \leq \frac{\exp(-\beta W_2^2(X_j, \xi))}{\exp(-\beta W_2^2(X_i, \xi))} \leq \exp(-\beta(D - 2\sqrt{D}r)) = \frac{\varepsilon}{N-1}\,. \tag{21}$$

Define $g(t) := \frac{e^{-\beta a_j(t)}}{\sum_{k=1}^N e^{-\beta a_k(t)}}$, where $a_j(t) := W_2^2(X_j, \xi_t)$ and $\xi_t = ((1-t)\mathsf{Id} + tT)_\# \xi_1$ be the geodesic interpolation of $\xi_1, \xi_2$ in the Wasserstein space, where $T$ is the optimal transport map from $\xi_1$ to $\xi_2$. By the mean-value theorem, there exists a $t^* \in [0, 1]$ such that:

$$w_j(\xi_2) - w_j(\xi_1) = g(1) - g(0) = g'(t^*)\,. \tag{22}$$

Note that

$$
\begin{aligned}
g'(t) &= w_j(\xi_t)\beta\left[\sum_{k=1}^N w_k(\xi_t)a_k'(t) - a_j'(t)\right] \\
&= w_j(\xi_t)\beta\left[\sum_{k=1}^N w_k(\xi_t)(a_k'(t) - a_j'(t))\right]\,.
\end{aligned}
\tag{23}
$$

Next, for Gaussian measures $X_k = \mathcal{N}(\mu_k, \Sigma_k)$ and $\xi_t = \mathcal{N}(m_t, \Omega_t)$, by Lemma 1, we have:

$$a_k(t) = W_2^2(X_k, \xi_t) = \|\mu_k - m_t\|^2 + \mathsf{CovErr}(\Sigma_k, \Omega_t)\,.$$

So,

$$a_k'(t) = 2\langle \frac{d}{dt}m_t\,,\, m_t - \mu_k\rangle + \frac{d}{dt}\mathsf{CovErr}(\Sigma_k, \Omega_t)\,.$$

Since $\xi_t$ is a geodesic between $\xi_1$ and $\xi_2$, we have $m_t = (1-t)m_1 + tm_2$ and therefore $\frac{d}{dt}m_t = m_2 - m_1$. Hence,

$$a_k'(t) = 2\langle m_2 - m_1\,,\, m_t - \mu_k\rangle + \frac{d}{dt}\mathsf{CovErr}(\Sigma_k, \Omega_t)\,,$$

and

$$a'_k(t) - a'_j(t) = 2\langle m_2 - m_1, \mu_j - \mu_k \rangle + \frac{d}{dt}[\mathsf{CovErr}(\Sigma_k, \Omega_t) - \mathsf{CovErr}(\Sigma_j, \Omega_t)]. \qquad (24)$$

By plugging equation 24 into the expression for $g'(t)$ in equation 23, we obtain:

$$g'(t) = w_j(\xi_t)\beta \sum_{k=1}^{N} w_k(\xi_t) \left[ 2\langle m_2 - m_1, \mu_j - \mu_k \rangle + \frac{d}{dt}[\mathsf{CovErr}(\Sigma_k, \Omega_t) - \mathsf{CovErr}(\Sigma_j, \Omega_t)] \right]$$

$$= 2w_j(\xi_t)\beta\langle m_2 - m_1, \mu_j - \sum_{k=1}^{N} w_k(\xi_t)\mu_k \rangle + w_j(\xi_t)\beta \sum_{k=1}^{N} w_k(\xi_t)\frac{d}{dt}[\mathsf{CovErr}(\Sigma_k, \Omega_t) - \mathsf{CovErr}(\Sigma_j, \Omega_t)]$$

$$= 2w_j(\xi_t)\beta\langle m_2 - m_1, \mu_j - m'_t \rangle + w_j(\xi_t)\beta \sum_{k=1}^{N} w_k(\xi_t)\frac{d}{dt}[\mathsf{CovErr}(\Sigma_k, \Omega_t) - \mathsf{CovErr}(\Sigma_j, \Omega_t)],$$

$$(25)$$

where $m'_t = \sum_{k=1}^{N} w_k(\xi_t)\mu_k$.

Next, since $\xi_1, \xi_2 \in B_i$, by geodesic convexity, $\xi_{t^*} \in B_i$. This means by equation 19, equation 21, we have that for $j \neq i$:

$$w_i(\xi_{t^*}) \geq 1 - \varepsilon, \ w_j(\xi_{t^*}) \leq \frac{\varepsilon}{N-1}, \qquad (26)$$

Therefore, from equation 25 and equation 26, we obtain the following bound on difference in weights, for $j \neq i$:

$$w_j(\xi_2) - w_j(\xi_1) = g'(t^*)$$

$$= 2w_j(\xi_{t^*})\beta\langle m_2 - m_1, \mu_j - m'_{t^*} \rangle + w_j(\xi_{t^*})\beta \sum_{k=1}^{N} w_k(\xi_{t^*})\frac{d}{dt}[\mathsf{CovErr}(\Sigma_k, \Omega_t) - \mathsf{CovErr}(\Sigma_j, \Omega_t)] \Big|_{t=t^*}$$

$$\leq \frac{2\beta\varepsilon}{N-1}\langle m_2 - m_1, \mu_j - m'_{t^*} \rangle + \frac{\varepsilon^2\beta}{(N-1)^2} \sum_{k \neq i} \frac{d}{dt}[\mathsf{CovErr}(\Sigma_k, \Omega_t) - \mathsf{CovErr}(\Sigma_j, \Omega_t)] \Big|_{t=t^*}$$

$$+ \frac{\varepsilon\beta}{N-1} w_i(\xi_{t^*})\frac{d}{dt}[\mathsf{CovErr}(\Sigma_i, \Omega_t) - \mathsf{CovErr}(\Sigma_j, \Omega_t)] \Big|_{t=t^*}. \qquad (27)$$

Next, we prove upper bounds for each term in equation 27. By Cauchy-Schwarz inequality:

$$|\langle m_2 - m_1, \mu_j - m'_{t^*} \rangle| \leq \|m_2 - m_1\| \cdot \|\mu_j - m'_{t^*}\|. \qquad (28)$$

Next, by the definition of $m'_{t^*}$ and the bounds on weights in equation 26, we get:

$$\|\mu_j - m'_{t^*}\| = \Big\| \mu_j - w_i(\xi_{t^*})\mu_i - \sum_{k \neq i} w_k(\xi_{t^*})\mu_k \Big\|$$

$$\leq \|w_i(\xi_{t^*})(\mu_j - \mu_i)\| + \Big\| (1 - w_i(\xi_{t^*}))\mu_j - \sum_{k \neq i} w_k(\xi_{t^*})\mu_k \Big\|$$

$$\leq (1 - \varepsilon)2M_W + \varepsilon M_W + (N-1) \cdot \frac{\varepsilon}{N-1} M_W$$

$$\leq 2M_W. \qquad (29)$$

By plugging equation 29 into equation 28, we obtain:

$$|\langle m_2 - m_1, \mu_j - m'_{t^*} \rangle| \leq 2M_W \|m_2 - m_1\| \leq 2M_W W_2(\xi_1, \xi_2). \qquad (30)$$

Next, we bound the second and third terms in equation 27. Since $\Sigma_k, \Omega_t$ commute by Lemma 4, they share a common eigenbasis so let $\Sigma_k = U\Lambda_{\Sigma_k}U^T$ and $\Omega_t = U\Lambda_{\Omega_t}U^T$ where $\Lambda_{\Sigma_k} = \mathsf{diag}(\sigma_{k,1}, \sigma_{k,2}, \ldots, \sigma_{k,d})$ and $\Lambda_{\Omega_t} = \mathsf{diag}(\omega_1(t), \omega_2(t), \ldots, \omega_d(t))$. Therefore, from equation 15 and definition of $\mathsf{CovErr}$, we obtain:

$$\frac{d}{dt}\mathsf{CovErr}(\Sigma_k, \Omega_t) = \sum_{i=1}^{d} 2(\sqrt{\sigma_{k,i}} - \sqrt{\omega_i(t)}) \cdot \left( -\frac{1}{2\sqrt{\omega_i(t)}} \right) \omega'_i(t),$$

and

$$\frac{d}{dt}[\mathsf{CovErr}(\Sigma_k, \Omega_t) - \mathsf{CovErr}(\Sigma_j, \Omega_t)] = -\sum_{i=1}^{d} \frac{\sqrt{\sigma_{k,i}} - \sqrt{\sigma_{j,i}}}{\sqrt{\omega_i(t)}} \omega_i'(t).$$

Next, by Cauchy-Schwarz inequality, the equation equation 15, by and the definition of $M_W$, we get:

$$\left| \frac{d}{dt}[\mathsf{CovErr}(\Sigma_k, \Omega_t) - \mathsf{CovErr}(\Sigma_j, \Omega_t)] \right| \leq \sqrt{\sum_{i=1}^{d}(\sqrt{\sigma_{k,i}} - \sqrt{\sigma_{j,i}})^2} \cdot \sqrt{\sum_{i=1}^{d} \frac{\omega_i'(t)^2}{\omega_i(t)}}$$

$$= \sqrt{\mathsf{CovErr}(\Sigma_k, \Sigma_j)} \cdot \sqrt{\sum_{i=1}^{d} \frac{\omega_i'(t)^2}{\omega_i(t)}}$$

$$\leq W_2(X_k, X_j) \cdot \sqrt{\sum_{i=1}^{d} \frac{\omega_i'(t)^2}{\omega_i(t)}}$$

$$\leq 2M_W \cdot \sqrt{\sum_{i=1}^{d} \frac{\omega_i'(t)^2}{\omega_i(t)}}. \tag{31}$$

By Lemma 4, we obtain
$$\omega_i'(t) = 2((1-t) + t\tau_i)(\tau_i - 1)\omega_{1,i},$$
where $\tau_i = \sqrt{\omega_{2,i}/\omega_{1,i}}$. By plugging the above in equation 31, we get:

$$\sum_{i=1}^{d} \frac{\omega_i'(t)^2}{\omega_i(t)} = \frac{4((1-t) + t\tau_i)^2(\tau_i - 1)^2\omega_{1,i}^2}{((1-t) + t\tau_i)^2 \omega_{1,i}}$$

$$= \sum_{i=1}^{d} 4(\tau_i - 1)^2\omega_{1,i}$$

$$= \sum_{i=1}^{d} 4(\sqrt{\omega_{2,i}} - \sqrt{\omega_{1,i}})^2$$

$$= 4\mathsf{BW}^2(\Omega_1, \Omega_2)$$

$$\leq 4W_2^2(\xi_1, \xi_2). \tag{32}$$

By plugging equation 32 into equation 31 yields:

$$\left| \frac{d}{dt}[\mathsf{BW}^2(\Sigma_k, \Omega_t) - \mathsf{BW}^2(\Sigma_j, \Omega_t)] \right| \leq 4M_W W_2(\xi_1, \xi_2).$$

Therefore, the second term in equation 27 is bounded by

$$\frac{\varepsilon^2\beta}{(N-1)^2} \left| \sum_{k \neq i} \frac{d}{dt}[\mathsf{BW}^2(\Sigma_k, \Omega_t) - \mathsf{BW}^2(\Sigma_j, \Omega_t)] \right|_{t=t^*} \leq \frac{\varepsilon^2\beta}{N-1} 4M_W W_2(\xi_1, \xi_2), \tag{33}$$

and the third term in equation 27 is bounded by

$$\frac{\varepsilon\beta}{N-1} w_i(\xi_{t^*}) \frac{d}{dt}[\mathsf{CovErr}(\Sigma_i, \Omega_t) - \mathsf{CovErr}(\Sigma_j, \Omega_t)] \bigg|_{t=t^*} \leq \frac{\varepsilon\beta}{N-1} 4M_W W_2(\xi_1, \xi_2). \tag{34}$$

Next, we plug the bounds in equation 30, equation 33, equation 34 into equation 27 to obtain for $j \neq i$:

$$|w_j(\xi_1) - w_j(\xi_2)| \leq \frac{2\beta\varepsilon}{N-1} \cdot 2M_W W_2(\xi_1, \xi_2) + \frac{\varepsilon^2\beta}{N} 4M_W W_2(\xi_1, \xi_2) + \frac{\varepsilon\beta}{N-1} 4M_W W_2(\xi_1, \xi_2)$$

$$= \frac{4\beta\varepsilon M_W(2 + \varepsilon) W_2(\xi_1, \xi_2)}{N-1}. \tag{35}$$

The final steps of the proof provide upper bounds for the two terms in equation 17 using the bound on $|\Delta w_j|$ in equation 35 and, therefore, prove contraction. We start with the first term in equation 17. By using $\|\mu_j - \mu_i\| \leq 2M_W$ and the bound in equation 35, we get

$$\left\| \sum_{j=1}^{N} \Delta w_j \mu_j \right\| = \| \sum_{j \neq i} (w_j(\xi_1) - w_j(\xi_2))(\mu_j - \mu_i)\|$$

$$\leq \sum_{j \neq i} |w_j(\xi_1) - w_j(\xi_2)| \|\mu_j - \mu_i\|$$

$$\leq 8\beta\varepsilon M_W^2 (2 + \varepsilon) W_2(\xi_1, \xi_2). \tag{36}$$

Next, we provide an upper bound on the second term in equation 17. Using $\lambda_{\min} \leq \lambda_{j,\ell} \leq \lambda_{\max}$, for all $j, \ell$, and the bound in equation 35, we get

$$\sum_{\ell=1}^{d} \left( \sum_{j=1}^{N} \Delta w_j \sqrt{\lambda_{j,\ell}} \right)^2 = \sum_{\ell=1}^{d} \left( \sum_{j \neq i} \Delta w_j (\sqrt{\lambda_{j,\ell}} - \sqrt{\lambda_{i,\ell}}) \right)^2$$

$$\leq d(\sqrt{\lambda_{\max}} - \sqrt{\lambda_{\min}})^2 16\beta^2 \varepsilon^2 M_W^2 (2 + \varepsilon)^2 W_2^2(\xi_1, \xi_2). \tag{37}$$

By plugging the bounds in equation 36 and equation 37 into equation 17, we obtain

$$W_2^2(\Phi(\xi_1), \Phi(\xi_2)) \leq 16\beta^2 \varepsilon^2 (2 + \varepsilon)^2 M_W^2 (4M_W^2 + d(\sqrt{\lambda_{\max}} - \sqrt{\lambda_{\min}})^2) W_2^2(\xi_1, \xi_2). \tag{38}$$

Next, we provide an upper bound for $\varepsilon$. From equation Assumption: Separation, the definition $D := 4\lambda_{\min} \left( \Delta_i - \frac{d}{2} \log(4\pi\lambda_{\max}) \right)$, and equation Assumption: Constraint, we get

$$D \geq \frac{4}{\beta} \log \left( \left(4M_W^2 + 2d(\lambda_{\max} + \lambda_{\min})\right) N^3 \beta \right) > \frac{8}{\beta} > \frac{1}{\beta N}, \tag{39}$$

for all $N \geq 1$. To simplify notation, define $L := \frac{1}{2\beta} \log \left( \left(4M_W^2 + 2d(\lambda_{\max} + \lambda_{\min})\right) N^3 \beta \right)$. From equation 39, we have $D \geq 8L$. Define $h(x) := -x + 2\sqrt{x}r$, where $r := \frac{1}{\sqrt{\beta N}}$. Note that $h'(x) < 0$ when $x > \frac{1}{\beta N}$. This means $h(D) \leq h(8L)$ for all $D \geq 8L$. Next, we show that $h(8L) < L$. Since $h(8L) = -8L + 4\frac{\sqrt{2L}}{\sqrt{\beta N}}$, proving $h(8L) < L$ is equivalent to showing $\frac{4\sqrt{2L}}{\sqrt{\beta N}} < 9L$. By plugging in the value of $L$, the previous inequality is equivalent to showing $\frac{64}{N} < 81 \log \left( \left(4M_W^2 + 2d(\lambda_{\max} + \lambda_{\min})\right) N^3 \beta \right)$, which is true for all $N \geq 1$ and equation Assumption: Constraint.

Next, since $h(8L) = -D + 2\sqrt{D}r < L$, by the definition of $\varepsilon$ in equation 20, the definition of $L$ in the above paragraph, we obtain

$$\varepsilon = (N - 1) \exp\left( -\beta(D - 2\sqrt{D}r) \right)$$

$$\leq (N - 1) \exp(-\beta L)$$

$$= (N - 1) \exp\left( -\frac{1}{2\beta} \log \left( \left(4M_W^2 + 2d(\lambda_{\max} + \lambda_{\min})\right) N^3 \beta \right) \right)$$

$$< \frac{N}{\sqrt{\left(4M_W^2 + 2d(\lambda_{\max} + \lambda_{\min})\right) N^3 \beta}}. \tag{40}$$

Define $C := \left(4M_W^2 + 2d(\lambda_{\max} + \lambda_{\min})\right)$. From equation 40, we have $\varepsilon < 1$ for $N > \frac{1}{C\beta}$. By using this fact, along with plugging equation 40 into equation 38, we obtain

$$W_2^2(\Phi(\xi_1), \Phi(\xi_2)) < \frac{144\beta M_W^2}{N} W_2^2(\xi_1, \xi_2). \tag{41}$$

Finally, since $N > 144\beta M_W^2$, we get $W_2(\Phi(\xi_1), \Phi(\xi_2)) < \kappa W_2(\xi_1, \xi_2)$, where $\kappa = \frac{144\beta M_W^2}{N}$, which finishes the proof of this lemma. $\qquad\square$

The next lemma uses Lemma 3 and 5 to apply Banach's fixed point theorem and show the existence of a unique fixed point in a neighborhood of each pattern.

**Lemma 6.** *Let $X_i = \mathcal{N}(\mu_i, \Sigma_i)$ for $i = 1, 2, \ldots, N$ be the patterns, where $\mu_i \in \mathbb{R}^d$ and $\Sigma_i \succ 0$. Define the Wasserstein ball $B_i := \{\nu \in \mathcal{P}_2(\mathbb{R}^d), W_2(X_i, \nu) \leq r\}$, where $r = \frac{1}{\sqrt{\beta N}}$. Under the assumptions of Lemmas 3 and 5, the map $\Phi$ has a unique fixed point in $B_i$.*

*Proof of Lemma 6.* First, we show that $B_i$ is closed in $(\mathcal{P}_2(\mathbb{R}^d), W_2)$. Let $\{\nu_n\}_{n \geq 1}$ be a sequence of probability measures in $B_i$ that converge to $\nu \in \mathcal{P}_2(\mathbb{R}^d)$ in $W_2$-distance. Since $\nu_n \in B_i$ for all $n$, we have $W_2(X_i, \nu_n) \leq r_i$. For any $n$, by triangle inequality:

$$W_2(X_i, \nu_n) \leq W_2(X_i, \nu) + W_2(\nu, \nu_n).$$

Taking $\limsup$ as $n \to \infty$ and using $\limsup_{n \to \infty} W_2(\nu, \nu_n) = 0$, we get

$$\limsup_{n \to \infty} W_2(X_i, \nu_n) \leq W_2(X_i, \nu). \tag{42}$$

From another application of triangle inequality, we have $W_2(X_i, \nu) - W_2(\nu, \nu_n) \leq W_2(X_i, \nu_n)$. Taking $\liminf$ as $n \to \infty$, we get:

$$\liminf_{n \to \infty} W_2(X_i, \nu_n) \geq W_2(X_i, \nu). \tag{43}$$

Therefore, from equation 42 and equation 43, we obtain $\lim_{n \to \infty} W_2(X_i, \nu_n) = W_2(X_i, \nu)$ and that $B_i$ is closed. Since $(\mathcal{P}_2(\mathbb{R}^d), W_2)$ is a closed metric space and $B_i$ is a closed subset of $\mathcal{P}_2(\mathbb{R}^d)$, we have that $(B_i, W_2)$ is a complete metric space.

Finally, since $(B_i, W_2)$ is a complete metric space, $\Phi$ is a self-map in $B_i$ from Lemma 3, and $\Phi$ is a contraction mapping from Lemma 5, we get from Banach's fixed point theorem that $\Phi$ has a unique fixed point in $B_i$. $\qquad\square$

### 7.4 PROOF OF THEOREM 1

*Proof of Theorem 1.* First, we introduce Algorithm 3 which can be used to sample Gaussian measures from the Wasserstein sphere $\mathcal{S}_R$ whose covariance matrices commute pairwise and whose eigenvalues lie in a bounded interval $[\lambda_{\min}, \lambda_{\max}]$.

---

**Algorithm 3** Sampling from commuting Gaussian measures on Wasserstein sphere

**Require:** $R > 0$ (sphere radius), $N \in \mathbb{N}$ (number of samples), $\lambda_{\min}, \lambda_{\max} > 0$ (eigenvalue bounds with $\lambda_{\min} < \lambda_{\max}$), $d \in \mathbb{N}$ (dimension)
**Ensure:** $\{X_1, \ldots, X_N\}$ where $X_i \sim \mathcal{N}(\mu_i, \Sigma_i)$ on $\mathcal{S}_R$
 1: **Initialize:** Fix an orthogonal matrix $U \in \mathbb{R}^{d \times d}$
 2: Define polytope $P = \left\{ (\lambda_1, \ldots, \lambda_d) \in \mathbb{R}^d : \sum_{k=1}^d \lambda_k = \frac{R^2}{2}, \lambda_{\min} \leq \lambda_k \leq \lambda_{\max} \text{ for all } k \right\}$
 3: **for** $i = 1$ to $N$ **do**
 4: $\quad$ Sample $(\lambda_1^{(i)}, \ldots, \lambda_d^{(i)}) \sim \text{Uniform}(P)$
 5: $\quad$ Construct covariance matrix: $\Sigma_i \leftarrow U \cdot \text{diag}(\lambda_1^{(i)}, \ldots, \lambda_d^{(i)}) \cdot U^T$
 6: $\quad$ Sample mean vector: $\mu_i \sim \text{Uniform}\left( \left\{ \mu \in \mathbb{R}^d : \|\mu\|_2 = \frac{R}{\sqrt{2}} \right\} \right)$
 7: $\quad$ Set $X_i \leftarrow \mathcal{N}(\mu_i, \Sigma_i)$
 8: **end for**
 9: **return** $\{X_1, \ldots, X_N\}$

---

Next, note that the polytope $P$ is non-empty with $R = \sqrt{d(\lambda_{\max} + \lambda_{\min})}$ since $d\lambda_{\min} < \frac{R^2}{2} = \frac{d(\lambda_{\max} + \lambda_{\min})}{2} < d\lambda_{\max}$. Also, note that by definition of $M_W$, we have $M_W = \max_i W_2(\delta_0, X_i) = R$.

If $X_i = \mathcal{N}(\mu_i, \Sigma_i), X_j = \mathcal{N}(\mu_j, \Sigma_j)$, then by the definition of the $L^2$ inner product, since $\Sigma_i, \Sigma_j$ share a common eigenbasis $U$, and because all the eigenvalues lie in the interval $[\lambda_{\min}, \lambda_{\max}]$, we have almost surely

$$-\log\langle X_i, X_j \rangle_{L^2} = \frac{d}{2}\log(2\pi) + \frac{1}{2}\sum_{k=1}^d \log(\lambda_{i,k} + \lambda_{j,k}) + \frac{1}{2}\sum_{k=1}^d \frac{[U^T(\mu_i - \mu_j)]_k^2}{\lambda_{i,k} + \lambda_{j,k}}$$

$$\geq \frac{d}{2}\log(4\pi\lambda_{\min}) + \frac{1}{4\lambda_{\max}}\|\mu_i - \mu_j\|^2 \,. \tag{44}$$

From equation 44, we get that the separation condition in equation Assumption: Separation is satisfied if

$$\frac{d}{2}\log(4\pi\lambda_{\min}) + \min_{j\neq i}\frac{1}{4\lambda_{\max}}\|\mu_i - \mu_j\|^2 \geq \frac{d}{2}\log(4\pi\lambda_{\max}) + \frac{1}{\beta\lambda_{\min}}\log\left(CN^3\beta\right) \,, \tag{45}$$

where $C := 4M_W^2 + 2d(\lambda_{\max} + \lambda_{\min})$. Since $M_W^2 = R^2 = d(\lambda_{\max} + \lambda_{\min})$, we get $C = 6d(\lambda_{\max} + \lambda_{\min})$. The condition in equation 45 is equivalent to

$$\min_{j\neq i}\|\mu_i - \mu_j\|^2 \geq 2d\lambda_{\max}\log(\gamma) + \frac{4\gamma}{\beta}\log(CN^3\beta) =: T \,, \tag{46}$$

where $\gamma := \frac{\lambda_{\max}}{\lambda_{\min}}$. The next steps of the proof are to find the conditions under which equation 46 holds with high probability.

By Vershynin (2009)[Theorem 3.4.5], we have that for independent uniform random vectors $u, v$ on a unit sphere $\mathbb{S}^{d-1}$ and $t > 0$:

$$\mathbb{P}(\langle u, v \rangle > t) \leq 2\exp(-t^2 d/2) \,. \tag{47}$$

Since $\mu_i, \mu_j$ are independent uniform random vectors on a sphere of radius $\frac{R}{\sqrt{2}} = \sqrt{\frac{d(\lambda_{\max}+\lambda_{\min})}{2}}$, we have

$$\|\mu_i - \mu_j\|^2 = R^2 - 2\langle \mu_1, \mu_2 \rangle = d(\lambda_{\max} + \lambda_{\min})(1 - \langle u, v \rangle) \,, \tag{48}$$

where $u, v$ are independent uniform random vectors on the unit sphere $\mathbb{S}^{d-1}$. Therefore, from equation 47, equation 48, we get

$$\mathbb{P}(\|\mu_i - \mu_j\|^2 < T) = \mathbb{P}\left(\langle u, v \rangle > 1 - \frac{T}{d\lambda_{\max}}\right) \leq 2\exp\left(-\frac{d}{2}\left(1 - \frac{T}{d\lambda_{\max}}\right)^2\right) \,. \tag{49}$$

By plugging in $N = \sqrt{\frac{p}{2}}\exp\left(\frac{d\alpha^2}{16}\right)$ and using $C = 6d(\lambda_{\max} + \lambda_{\min})$, we get

$$\frac{4}{d\lambda_{\min}\beta}\log(CN^3\beta) = \frac{4}{d\lambda_{\min}\beta}\left(3\log N + \log\beta + \log C\right)$$

$$= \frac{4}{d\lambda_{\min}\beta}\left(\frac{3}{2}\log\left(\frac{p}{2}\right) + \frac{3d\alpha^2}{16} + \log\beta + \log(2d(3\lambda_{\max} + \lambda_{\min}))\right)$$

$$= \frac{3\alpha^2}{4\beta\lambda_{\min}} + \frac{6\log(p/2) + 4\log(\beta) + 4\log(6d(\lambda_{\max} + \lambda_{\min}))}{d\beta\lambda_{\min}} \,. \tag{50}$$

From the fact that $\alpha > 0$ from the statement of the theorem, it follows that if $\beta > \frac{3\alpha}{\lambda_{\min}}$, then the first term in equation 50 satisfies $\frac{3\alpha^2}{4\beta\lambda_{\min}} < \frac{\alpha}{4}$, and there exists $d_0^{(1)} \in \mathbb{N}$ such that the second term in equation 50 can be bounded as

$$\frac{6\log(p/2) + 4\log(\beta) + 4\log(6d(\lambda_{\max} + \lambda_{\min}))}{d\beta\lambda_{\min}} < \frac{\alpha}{4} \,.$$

Therefore,

$$\frac{4}{d\lambda_{\min}\beta}\log(CN^3\beta) < \frac{\alpha}{2} \,. \tag{51}$$

From equation 51 and the definition of $T$ in equation 46, we obtain

$$\frac{T}{d\lambda_{\max}} < 2\log(\gamma) + \frac{\alpha}{2} = 1 - \alpha + \frac{\alpha}{2} = 1 - \frac{\alpha}{2} \,. \tag{52}$$

By plugging equation 52 into equation 49, we get

$$\mathbb{P}(\|\mu_i - \mu_j\|^2 < T) \leq 2\exp\left(-\frac{d\alpha^2}{8}\right) \tag{53}$$

By applying union bound to all the pairs $(i, j)$ in equation 53, and using the definition of $N = \sqrt{\frac{p}{2}} \exp\left(\frac{d\alpha^2}{8}\right)$, we obtain

$$\mathbb{P}\left(\exists i, j \text{ such that } \|\mu_i - \mu_j\|^2 < T\right) < \frac{N^2}{2} \cdot 2 \exp\left(-\frac{d\alpha^2}{8}\right) = \frac{p}{2} < p\,. \tag{54}$$

Finally,

$$\mathbb{P}\left(\min_{j \neq i} \|\mu_i - \mu_j\|^2 < T\right) \leq \mathbb{P}\left(\exists i, j \text{ such that } \|\mu_i - \mu_j\|^2 < T\right) < p\,,$$

which means the condition in equation 46 holds with probability $1 - p$ and this finishes the proof of the statement that we can sample $N = \sqrt{\frac{p}{2}} \exp\left(\frac{d\alpha^2}{16}\right)$ patterns which are separated in $L^2$-inner product by the condition described in equation Assumption: Separation.

Since the separation condition in equation Assumption: Separation is satisfied for $N = \sqrt{\frac{p}{2}} \exp\left(\frac{d\alpha^2}{16}\right)$ random patterns with probability $1 - p$ for $d > d_0^{(1)}$, because the constraint on $\beta$ in equation Assumption: Constraint is satisfied from the assumption of this theorem $\beta > \frac{3\alpha}{\lambda_{\min}}$ for large enough $d_0^{(2)}$, and because

$$N = \sqrt{\frac{p}{2}} \exp\left(\frac{d\alpha^2}{16}\right) > 144\beta M_W^2 = 144\beta R^2 = 144\beta(\lambda_{\min} + \lambda_{\max})$$

for large enough $d_0^{(3)}$, it follows from Lemma 5 that $\Phi$ is a contractive mapping with probability $1 - p$ for $N = \sqrt{\frac{p}{2}} \exp\left(\frac{d\alpha^2}{16}\right)$ random patterns and $d > \max\{d_0^{(1)}, d_0^{(2)}, d_0^{(3)}\}$. Finally, it follows from Lemma 6 that if we define $B_i = \{\nu \in \mathcal{P}_2(\mathbb{R}^d)\,, W_2(X_i, \nu) < \frac{1}{\sqrt{\beta N}}\}$, then there exists a unique fixed point in $B_i$ for all $i \in \{1, 2, \ldots, N\}$. Hence, from Definition 1, we get that $N = \sqrt{\frac{p}{2}} \exp\left(\frac{d\alpha^2}{16}\right)$ random patterns can be stored on a Wasserstein sphere with probability $1 - p$ for all $p > 0$. $\qquad\square$

### 7.5 PROOF OF THEOREM 2

*Proof of Theorem 2.* Since the assumptions of Lemma 3 and Lemma 5 hold, the existence of a unique fixed point follows from Lemma 6. Next, by Lemma 5, we get $W_2(\Phi(\xi), X_i^*) < \kappa$ because $\xi, X_i^* \in B_i$, where $\kappa$ is the contraction coefficient from Lemma 5. Applying this inequality iteratively, we obtain

$$W_2(\Phi^n(\xi), X_i^*) \leq \kappa^n W_2(\xi, X_i^*) \leq \kappa^n (2r)\,. \tag{55}$$

By setting $\kappa^n(2r) = \varepsilon$, using the definition $\kappa = \frac{144\beta M_W^2}{N}$ from Lemma 5, the definition $r = \frac{1}{\sqrt{\beta N}}$, and noting that $\varepsilon < r$ finishes the proof of this theorem. $\qquad\square$

### 7.6 PROOF OF THEOREM 3

*Proof of Theorem 3.* By triangle inequality, using the fact that the unique fixed point $X_i^* \in B_i$, appealing to $\Phi(X_i^*) = X_i^*$ since $X_i^*$ is a fixed point, and applying Lemma 5, we obtain

$$\begin{aligned} W_2(\Phi(\xi), X_i) &\leq W_2(\Phi(\xi), X_i^*) + W_2(X_i, X_i^*) \\ &\leq W_2(\Phi(\xi), \Phi(X_i^*)) + \frac{1}{\sqrt{\beta N}} \\ &\leq W_2(\xi, X_i^*) + \frac{1}{\sqrt{\beta N}} \\ &\leq \frac{3}{\sqrt{\beta N}}\,. \end{aligned} \tag{56}$$

$\qquad\square$

