# OpenReview forum: "Dense Associative Memory on the Bures-Wasserstein Space"
_ICLR.cc/2026/Conference — ICLR 2026 Conference Withdrawn Submission_

### Official Review · Reviewer_YN4m · 2025-10-21

**Soundness:** 3
**Presentation:** 3
**Contribution:** 2
**Rating:** 2
**Confidence:** 3

**Summary:**

This paper proposes a novel generalization of dense associative memories (DAMs) from Euclidean vector spaces to the Bures–Wasserstein manifold of Gaussian probability distributions. Traditional DAMs operate over vector representations with fixed energy functions, achieving exponential storage capacity and robust retrieval. However, they are limited to deterministic vectorial patterns. This work extends the DAM framework to the space of Gaussian measures, enabling storage and retrieval of full probability distributions rather than point embeddings.

**Strengths:**

$\textbf{Novel conceptual contribution:}$

The paper introduces the first principled extension of dense associative memories to the Wasserstein space, bridging two distinct research areas: associative memory models and optimal transport geometry.

$\textbf{Rigorous theoretical development:}$

Τhe authors employ very interesting mathematical tools to provide a thorough and well-structured analysis. Their proofs rigorously support the storage capacity and retrieval guarantees, demonstrating careful reasoning in extending associative memories to the Wasserstein space of Gaussian distributions.

$\textbf{Empirical resuls:}$

Experiments validate the theoretical findings, showing phase transitions in convergence behavior and sharp energy basins both on synthetic Gaussian data and real Gaussian embeddings.

**Weaknesses:**

$\textbf{Strong simplifying assumptions:}$

The main theorems assume pairwise-commuting covariance matrices, which substantially simplifies the geometry and may not hold for realistic distributional data. Assumption 1 also imposes strong separation between stored Gaussians, requiring exponentially small overlap in dimension $d$. While this ensures stable retrieval and enables rigorous theoretical guarantees, the authors address the feasibility of these assumptions by sampling synthetic Gaussians from a high-dimensional Wasserstein sphere. This controlled setup makes the analysis possible, but it remains idealized and may not fully reflect real-world distributions. I remain unconvinced regarding this assumption.

$\textbf{Limited empirical diversity:}$


While experiments on synthetic Gaussians and Gaussian word embeddings are appropriate, they remain narrow in scope. Demonstrations on more complex or high-dimensional generative tasks would strengthen the practical relevance.

$\textbf{Computational efficiency and scalability:}$

The paper does not discuss the computational cost of iterative Wasserstein updates in high dimensions, which could be nontrivial given the need for matrix square roots and eigendecompositions.

**Questions:**

1) The main theoretical results (e.g., Theorem 1 and Theorem 2) assume pairwise-commuting covariance matrices. Can the authors clarify how critical this assumption is for their results? Could the analysis be partially extended to approximately commuting covariances? If not, could the authors justify why this assumption is reasonable?

2) The theoretical guarantees rely on exponentially small overlap between stored Gaussians in $d$, which ensures clean attractor basins but may be unrealistic. Can the authors quantify how sensitive the retrieval performance is to violations of this assumption? Do the empirical experiments suggest that some degree of overlap can be tolerated without loss of convergence or stability? Intuitively, very well-separated Gaussians seem easy to retrieve, but it is unclear how the method performs when overlaps are larger, as may happen in practice.

3) Each retrieval step requires $N$ Bures–Wasserstein distance computations and $d\times d$ matrix square roots. The computational cost of this seems large, especially as $d$ and $N$ starts to become larger.

4) Assuming all your assumptions hold in a practical scenario, how do you find the optimal $\beta$?

---

### Official Review · Reviewer_7Cw1 · 2025-10-30

**Soundness:** 2
**Presentation:** 2
**Contribution:** 2
**Rating:** 4
**Confidence:** 3

**Summary:**

The paper introduces an energy-based associative memory model for multi-variate Gaussian distributions. Its claimed contributions are:
- A novel energy formulation based on 2-Wasserstein distance between probability distributions
- Proof of exponential capacity with regard to the dimensionality
- Bounds on the fidelity of retrieval under noisy query distributions
- Empirical validation on simulated and real data

**Strengths:**

- The problem is well motivated
- The approach is novel. Even though I question later whether this could be equivalent to a simple MCHN, there is still some novelty in trying to apply MCHNs to probability distributions.
- The paper is clear
- The empirical results are reproducible

**Weaknesses:**

- One of the stated motivation is to be able to retrieve multi-modal distributions, however, the proposed methods only work for Gaussian distributions
- Supposing that all distributions commute further restrains the scope of potential applications
- The experiments lack comparison against other methods. I understand that there might not be other works addressing the same problem, but we could for instance imagine comparing against other associative memory models that try to retrieve stored Gaussian distributions using the vector of means $\mu_i$, or vector concatenations of means and eigenvalues  $[\mu_i, \sigma_i]$
- There seems to be issues in the proofs (cf. my questions).

**Questions:**

My main question is with regard to the actual novelty of the algorithm. It is unclear why using $\Phi$ on $\mathcal{N}(\mu,diag(\sigma\_i)\_i) \in \mathcal S_R$ is any different than using the MCHN's (Ramsauer et al. 2020) updating function on the concatenation $[\\mu,\\sigma] \\in \\mathbb R^{2d}$. The constraint that all the covariance matrices commute imposes that they are only described by the vector of eigenvalues as highlighted many times in the appendix. Then for Gaussian distribution $X_i=\\mathcal N(\mu_i,diag(\\sigma_{i,j}^2)_j)$ (as sampled in the notebook) and denoting $x_i=[\mu_i,\sigma_i]\in\mathbb R^{2d}$, we have $W_2^2(X_i,X_j)=\|\mu_i-\mu_j\|^2+\|\sigma_i-\sigma_j\|^2=\|x_i-x_j\|^2$ from Lemma 1. For $\xi=\mathcal N(m,diag(\gamma_j^2)_j)$ and denoting $y=[m,\gamma]$ we have $\Phi(\xi)=\mathcal N(\sum_i w_i(\xi)m_i,(\sum_iw_i(\xi)\sigma_i)^2 I)$ from Lemma 2. Thefore the updating function is $m'\leftarrow \sum_i w_i(\xi)\mu_i $ and $\gamma'\leftarrow \sum_iw_i(\xi)\sigma_i$ and therefore $y'\leftarrow\sum_i w_i(\xi)x_i$. Moreover

$$
\begin{split}
    w_i(\xi)&=\frac{\exp \left(-\beta W_2^2\left(X_i, \xi\right)\right)}{\sum_j \exp \left(-\beta W_2^2\left(X_j, \xi\right)\right)}\\
    &=\frac{\exp \left(-\beta \|x_i-y\|^2\right)}{\sum_j \exp \left(-\beta \| x_j-y\|^2\right)}\\
    &=\frac{\exp\left(-\beta\|x_i\|^2\right)
       \exp\left(-\beta\|y\|^2\right)
       \exp\left(2\beta x_i^\top y\right)}
       {\sum_j \exp\left(-\beta\|x_j\|^2\right)
       \exp\left(-\beta\|y\|^2\right)
       \exp\left(2\beta x_j^\top y\right)}\\
    &=\frac{\exp\left(2\beta x_i^\top y\right)}
    {\sum_j \exp\left(2\beta x_j^\top y\right)}
\end{split}
$$

The last line comes from $\forall i,\ \|x_i\|^2=\sum_j\mu_{i,j}^2+\sum_j \sigma_{i,j}^2=W_2^2(\delta_0,X_i)=R^2$ as $X_i\in\mathcal S_R$. Finally the updating function seems exactly the same introduced in MCHNs. To confirm it I ran an experiment in the notebook by using the mean updating function on $[\mu,\sigma]$ and it retrieves the same numerical results.

Therefore, in the case of commutating covariance matrices, why working in $\mathcal S_R$ with constraints on $\sigma$ instead of computing the eigenvalues of the covariance matrices then working in $\mathbb R^{2d}$ with MCHNs' updating function ? In the case of non-commutative covariance matrices, does this method achieve better results than working in $\mathbb R^{d+d(d+1)/2}$ with $ [\mu,\Sigma]$ despite the lack of theoretical guarantees ?

A second question is with the choice of constraining $X_i$ to be on a sphere. It looks necessary for MCHNs because their energy function uses a dot-product instead of a L2 norm. But in the case of the 2-Wasserstein distance, it doesn't look necessary. In figure 1 and figure 2, the stored patterns are not on a sphere. Do you fall back to the sphere setting only to obtain theoretical guarantees ?
At first glance, the storage capacity would improve if not constraining ourselves to the sphere, so this constraint really surprises me.

Other questions:
- You cite (Krotov and Hopfield, 2016) for the log sum exp energy function, but I couldn't find it in the paper. Can you point the exact equation in the paper ? Or maybe it comes from another article ? Maybe (Ramsauer et al. 2020) ?
- You also seem to say that using a distance in the log sum exp energy function comes from (Krotov and Hopfield, 2016). I can't find this formulation neither in (Krotov and Hopfield, 2016) and (Ramsauer et al. 2020).
- In order to demonstrate the exponential capacity of the model, shouldn't you also prove that the balls do not overlap ? or that the fixed points $X_i^*$ are distinct ?

Crucial questions with regard to the proofs. These points should be answered and the proofs should be fixed (or clarified).
- Line 1079: the mean error upper bound was in $\epsilon$, not $\epsilon^2$. If this inequality is still valid, could you explain how ? it is not trivial to me.
- Line 1105: it looks to me that the assumption $CN^3 \beta > e^2$ does not allow to conclude to this inequality, isn't it the other way around ?
- Line 1339: $w_i(\xi) \geq 1 - \epsilon$ and not the other way around, so it looks like it can't be used to move from 1337 to 1339 ?
- Line 1443: shouldn't it be $\beta L$ instead of $- \beta L$ ?


Other minor comments:
- Line 968: missing $U$ ?
- Line 1089: $\epsilon^2 <$ instead of $\epsilon$ ?
- Equation 34: should add $| . |$ ?
- Lemma 5 demonstration mixes $BW^2$ and $CovErr$ notations
- Lemma 5 demonstration uses inconsistent $i$ notation (both summation index and index of the ball)
- Same remark for theorem 2
- Line 1403: $\frac{1}{N-1}$ instead of $\frac{1}{N}$ ?
- Line 1439: should be $h(D)$ not $h(8L)$ ?
- Line 1538: $d(\lambda_{max} - \lambda_{min})$ became $d(\lambda_{max})$, why ?
- Line 1545: should be $6d($ instead of $2d(3$ ?
- Font sizes in figures could be bigger so that we don't have to zoom in

---

### Official Review · Reviewer_up3P · 2025-11-05

**Soundness:** 3
**Presentation:** 3
**Contribution:** 4
**Rating:** 4
**Confidence:** 4

**Summary:**

This paper introduces a novel framework for Dense Associative Memories (DAMs) designed to store and retrieve probability distributions, rather than the traditional vector-based patterns. The authors extend the typical log-sum-exponential (LSE) energy function to operate on the Bures-Wasserstein space, which is suited for Gaussian distributions (and patterns sampled from them).

**Strengths:**

1. **Novelty.** The work provides an interesting and new take on DAM where instead of storing typical patterns $x_i \sim \mathcal{N}(\mu_i, \Sigma_i)$. What if the model stores $\{ \mathcal{N}(\mu_i, \Sigma_i) \}^N_{i = 1}$ or normal distributions instead?

2. **Concrete.** The paper proves that this model can store an exponential number of Gaussian distributions (see Theorem 1) and extends the known exponential capacity of vector-based DAMs to the much more complex Bures-Wasserstein space. Moreover, the work provides (although limited) experimentation on their introduced Wasserstein-LSE energy on synthetic data and Gaussian text-embedding (see [Vilnis and McCallum (2014)](https://arxiv.org/abs/1412.6623)).

3. **Open directions**. I think the new approach introduced in this work opens more avenues to explore, especially the extensions to the model's applicability.

**Weaknesses:**

1. **Limited Experimentation**. Although the work did state that they want to demonstrate a proof-of-concept on the storage of Gaussian distributions. This is certainly fair. But, what about storing information coming from VAEs? Certainly, those models learn some sort of Gaussian-based $\mu_\theta$ and $\Sigma_\theta$ right?

2. **Unclear usability**. Going back to image domain. It is unclear on the practicality of this approach (especially for natural data like images) since it relies on the assumption of Gaussian distributions.

3. **Scalability**. The retrieval algorithm (1) requires computing $N$ Wasserstein distances and $N$ transport map coefficients at every iteration. This is difficult to do when $N$ is very large. Nonetheless, it is best to note that the same complaint here can be said about typical DAMs.

Notes about **presentation**:
1. All of the figures' (x and y) labels should be much larger.
2. $\mathbf{Id}$ in Eq. (2) is undefined. It is an identity map, right? Please make it clearer.

**Questions:**

See above for some questions proposed in the **Weaknesses** section. But allow me ask some "crazy" questions here...

1. The paper's main theoretical guarantees rely on the assumption that all covariance matrices commute pairwise. This seems like a very strong assumption that is unlikely to hold in real-world data. Can we relax this assumption? What would be the consequences?

2. What about **spurious** patterns? I guess in this case they would be "hallucinated distributions." See this [paper](https://arxiv.org/abs/2406.09358) as it may inspire something useful for the paper. But it would be interesting for you to discuss this aspect in the paper, as it is the hallmark topic of Associative Memory.

My score is actually a **5**, to be frank.

---

### Note · Authors · 2025-11-18

I have read and agree with the venue's withdrawal policy on behalf of myself and my co-authors.